# VLM-Guided Adaptive Negative Prompting for Creative Generation

**Shelly Golan**
Technion,
Tel-Aviv University

**Yotam Nitzan**
Adobe Research

**Zongze Wu**
Adobe Research

**Or Patashnik**
Tel-Aviv University

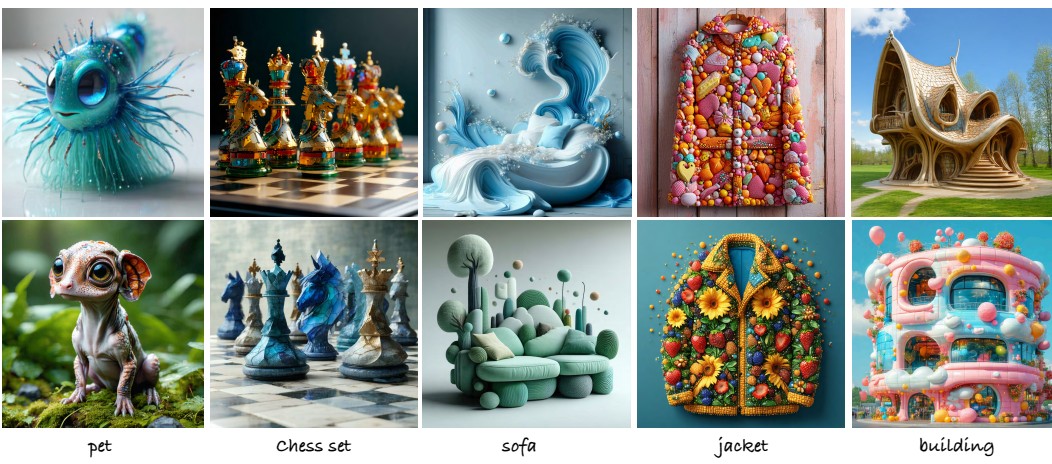

*"A photo of a creative ..."*

Figure 1: Our method generates creative concepts such as novel pets, uniquely designed jackets, and unconventional buildings by steering the generation away from conventional patterns using a VLM-Guided Adaptive Negative Prompting process.

## ABSTRACT

Creative generation is the synthesis of new, surprising, and valuable samples that reflect user intent yet cannot be envisioned in advance. This task aims to extend human imagination, enabling the discovery of visual concepts that exist in the unexplored spaces between familiar domains. While text-to-image diffusion models excel at rendering photorealistic scenes that faithfully match user prompts, they still struggle to generate genuinely novel content. Existing approaches to enhance generative creativity either rely on interpolation of image features, which restricts exploration to predefined categories, or require time-intensive procedures such as embedding optimization or model fine-tuning. We propose VLM-Guided Adaptive Negative-Prompting, a training-free, inference-time method that promotes creative image generation while preserving the validity of the generated object. Our approach utilizes a vision-language model (VLM) that analyzes intermediate outputs of the generation process and adaptively steers it away from conventional visual concepts, encouraging the emergence of novel and surprising outputs. We evaluate creativity through both novelty and validity, using statistical metrics in the CLIP embedding space. Through extensive experiments, we show consistent gains in creative novelty with negligible computational overhead. Moreover, unlike existing methods that primarily generate single objects, our approach extends to complex scenarios, such as generating coherent sets of creative objects and preserving creativity within elaborate compositional prompts. Our method integrates seamlessly into existing diffusion pipelines, offering a practical route to producing creative outputs that venture beyond the constraints of textual descriptions.

## 1 INTRODUCTION

A growing body of research (Hertzmann, 2018; Yongjun et al., 2025; Ivcevic & Grandinetti, 2024) revolves around a somewhat philosophical question: what are creativity and originality, and can

computers create art? One suggestion by Boden (2009) is to categorize computational creativity along a spectrum of increasing novelty. At the lowest level, *combinatorial* creativity produces unexpected combinations of existing concepts, such as a hybrid creature that merges features of a bee and a giraffe. *Exploratory* creativity goes further by discovering new possibilities within a known domain while maintaining validity, for instance, inventing an animal species with entirely new but biologically plausible traits. At the highest level, *transformational* creativity challenges the boundaries of existing categories altogether, such as conceiving an organism so unlike current life forms that it forces us to reconsider the definition of "animal" itself.

Recent advances in text-to-image (T2I) diffusion models have demonstrated strong capabilities in generating photorealistic images from natural language prompts. These models excel at reproducing and recombining simple visual concepts from their training data, allowing for combinatorial creativity to some extent. However, they still struggle with novelty that falls under the category of exploratory and transformational creativity. This limitation reflects an inherent tension in generative modeling between mode coverage (i.e., capturing the full distribution), and mode seeking (i.e., generating high-quality typical samples). For example, a known technique that attempts to navigate this tradeoff is Classifier-free guidance (CFG). Lower guidance scales increase diversity but compromise text alignment, while higher scales improve prompt adherence but generate more typical outputs.

Our experiments show that simple prompt modifications fail to produce creative outputs from current models. As demonstrated in Figure 2, adding creativity-related terms such as "creative" or "new type of" produces outputs that remain similar to conventional pets – like a blue cat with wings, kittens, dogs, or a ferret-like animal with long ears. On the other hand, our blue pet, presented in Figure 1, cannot be described as a combination of known pets.

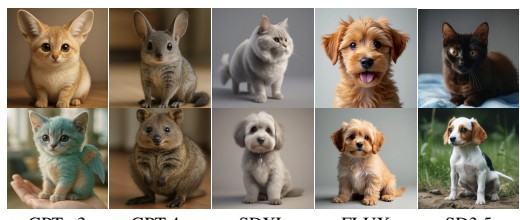

Figure 2: Images generated with GPT-o3 (OpenAI, 2025), GPT-4o (OpenAI, 2024), SDXL (Podell et al., 2023), FLUX-dev (Black Forest Labs, 2024), and SD3.5 (Esser et al., 2024) using the prompt "Professional high-quality photo of a new type of pet."

Existing frameworks for creative generation fall into two paradigms: *combinatorial* approaches that blend predefined concept pairs through rule-based searches (Li et al., 2024) or learnable tokens (Feng et al., 2024), and *exploratory* methods like ConceptLab (Richardson et al., 2024) that optimize textual embeddings to discover novel concepts. Specifically, ConceptLab formulates creative generation as an iterative optimization problem over a learned textual embedding, minimizing a loss function that balances two objectives: maintaining similarity to a broad target category while maximizing the distance from known subcategories in the CLIP embedding space. While these demonstrate progress, they require either per-concept optimization procedures, specialized training on curated datasets, or predefined concept specifications, limiting their practical deployment and scalability.

To address these limitations, we propose VLM-Guided Adaptive Negative-Prompting, a training-free method that integrates into any diffusion sampler without modifying pretrained weights or requiring curated datasets. Unlike previous approaches, our method operates entirely at inference time through a closed-loop feedback mechanism (Figure 3). We leverage a lightweight vision-language model (VLM) to adaptively steer the generation process away from its typical predictions and thus towards unexplored regions of possible outputs. Our approach utilizes the VLM to analyze intermediate denoising predictions at each timestep, identify dominant objects, and adaptively convert these observations into negative prompts that are integrated into the next denoising step.

Through experiments across multiple VLM models, diffusion pipelines, and human evaluation studies, we demonstrate consistent improvements in *exploratory* creativity while maintaining categorical coherence. Our analysis reveals how adaptive negative prompting guides the denoising trajectories toward unexplored semantic regions and highlights the importance of VLM feedback during inference. Through extensive ablation studies, we validate our key design choices, including dynamic negative prompt accumulation and per-generation adaptation, showing superiority over alternative approaches. Furthermore, we demonstrate capabilities beyond existing methods, including the generation of coherent creative sets and the preservation of creativity within complex compositional prompts, showcasing the versatility of our VLM-guided approach.

## 2 RELATED WORK

**Foundations of Creative Generation**    The pursuit of extending human imagination with machine learning has motivated extensive research in computational creativity, from algorithmic design tools (Cohen-Or & Zhang, 2016; Sims, 1994; 1991; Sun et al., 2025) to theoretical frameworks examining whether computers can create art or merely serve as sophisticated tools for human artists (Hertzmann, 2018). Early work, such as Xu et al. (2012), introduced a set-evolution framework for creative 3D shape modeling by steering the generation towards user-preferred shapes while maintaining diversity. Other works (Elgammal et al., 2017; Sbai et al., 2019) proposed modifying losses and training objectives to generate creative art by maximizing deviation from established styles while minimizing deviation from the general art distribution.

**Concept Blending and Combinatorial Creativity**    A significant portion of computational creativity involves combinatorial approaches. Some works (Liew et al., 2022; Zhou et al., 2025) leveraged diffusion models to blend different visual and semantic concepts for the generation of novel outputs. Dorfman et al. (2025) extended this to multiple visual inputs by crafting composite embeddings, stitched from the projections of multiple input images onto concept-specific CLIP-subspaces identified through text. For text-based concept pairs, Li et al. (2024) suggested balance swap-sampling, which generates creative combinatorial objects by randomly exchanging intrinsic elements of text embeddings and selecting high-quality combinations based on CLIP distances. Feng et al. (2024) takes a different approach and re-defines "creativity" as a learnable token. They iteratively sample diverse text pairs from their proposed dataset to form adaptive prompts and restrictive prompts, and then optimize the similarity between their respective text embeddings. While these combinatorial approaches recombine user-specified concepts, we instead discover novel concepts within broad categories without predefined targets.

**VLM-Guided Creativity Approaches.**    Recent research leverages Vision-Language Models (VLMs) to guide creative generation. Feng et al. (2025) uses VLMs to supervise distribution-conditional generation, enabling multi-class concept blending through a learnable encoder-decoder framework. While the above approaches focus on combinatorial creativity through concept blending, Richardson et al. (2024) introduces ConceptLab, which tackles the more challenging task of exploratory creativity. They formulate the Creative Text-to-Image (CT2I) generation as an optimization process of a learned textual embedding. To prevent convergence to existing concepts, ConceptLab incorporates a question-answering VLM that adaptively adds new constraints to the optimization problem. These VLM-guided approaches rely on per-concept optimization procedures that require multiple iterations and substantial computational resources. Our approach leverages VLMs as real-time oracles during the denoising process to reduce computational overhead.

**Optimization-Free Creative Generation.**    Han et al. (2025) boosts creativity in Stable Diffusion by amplifying features during denoising, primarily affecting color and textures. While we share the goal of optimization-free creativity enhancement, our method operates through dynamic negative prompting to guide the generation away from conventional semantic patterns rather than amplifying existing features. The advantage of such optimization-free approaches lies in their immediate applicability to existing models without requiring additional training or complex optimization procedures.

**Theory of Creative Generation**    Recent work has explored creative generation from a more theoretical point of view, investigating the relation between memorization and novel sample generation. Lu et al. (2024) propose a method which improves sample diversity and creativity of diffusion-based image generative models and to prevent training data reproduction. Shah et al. (2025) investigates whether creative generation requires memorization, proposing ambient diffusion techniques that reduce reliance on reproducing training data while maintaining generation quality. Kamb & Ganguli (2025) provides theoretical foundations by analyzing creativity mechanisms in convolutional diffusion models, offering formal frameworks for understanding how diffusion models can generate samples that do not exist in their training distributions.

## 3 METHOD

Our VLM-Guided Adaptive Negative-Prompting method enhances creative generation in diffusion models through a closed-loop feedback mechanism that dynamically navigates the denoising process away from familiar visual patterns. As illustrated in Figure 3, our method monitors the intermediate denoiser outputs using a Vision-Language Model (VLM), which identifies dominant elements (e.g.,

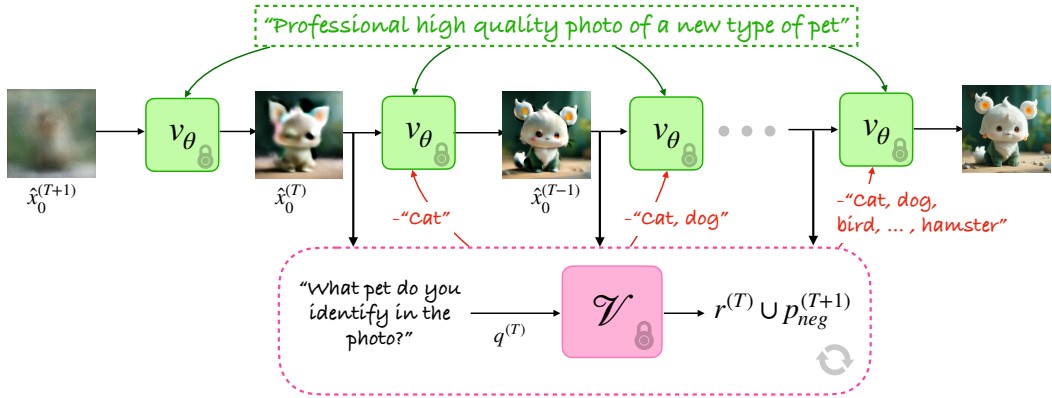

Figure 3: Overview of our VLM-guided negative prompting method. To generate a creative image (e.g., "new type of pet"), we sample Gaussian noise and perform an augmented denoising process that maintains an adaptive list of negative prompts. At each denoising step, we query a pre-trained Vision-Language Model (VLM) to identify visual concepts present in the intermediate output and update the list accordingly, steering the denoising process away from them. For example, we add the token "cat" to the accumulating list to shift the denoising trajectory away from generating an image resembling a cat as well as the previously detected pets.

"cat") and accumulates them as dynamic negative prompts during the generation process. This adaptive accumulation refines the guidance signal at each denoising step.

We begin by establishing the necessary background on negative prompting in Section 3.1 and detailing our VLM-guided synthesis strategy in Section 3.2.

## 3.1 BACKGROUND: DIFFUSION MODELS AND NEGATIVE PROMPTING

Diffusion models generate images by gradually denoising a sample from pure noise $x_T$ over a series of time steps. Latest diffusion models, including Stable Diffusion 3.5 (Esser et al., 2024) used in our experiments, employ flow matching (Lipman et al., 2023) to generate images through iterative denoising. Let $x_t$ denote the noisy image at timestep $t \in [T, ..., 0]$. In flow matching, the model learns a velocity field $v_\theta(x_t, t, c)$ conditioned on text embedding $c = E(p)$ derived from prompt $p$ via text encoder $E$. The denoising process follows the probability flow ODE: $\frac{dx_t}{dt} = v_\theta(x_t, t, c)$. During sampling, we can estimate the clean image at any timestep using the following equation:

$$\hat{x}_0^{(t)} = x_t - t \cdot v_\theta(x_t, t, c) \tag{1}$$

Classifier-free guidance (CFG) (Ho & Salimans, 2021) improves conditional generation by combining conditional and unconditional predictions: $\tilde{v}_\theta^w = v_\theta(x_t, t, \varnothing) + w \cdot (v_\theta(x_t, t, c) - v_\theta(x_t, t, \varnothing))$, where $\varnothing$ denotes the unconditional (null) embedding, and $w$ is the guidance scale. When $w = 0$, the model generates unconditional samples; as $w$ increases, the model increasingly favors features aligned with the conditioning text. The guidance operates by amplifying the difference between conditional and unconditional predictions. When $w = 0$, the model generates unconditional samples. As $w$ increases, the model increasingly favors features that align with the conditioning text. This mechanism was naturally extended (Saharia et al., 2022) to negative prompting, in which the model is explicitly discouraged from generating features associated with a negative prompt $p_{neg}$. Instead of subtracting the unconditional prediction, we subtract a negatively conditioned prediction:

$$\hat{v}_\theta^w = v_\theta(x_t, t, c_{neg}) + w \cdot (v_\theta(x_t, t, c_{pos}) - v_\theta(x_t, t, c_{neg})), \tag{2}$$

where $c_{neg} = E(p_{neg})$ represents the negative prompt embedding derived from the unwanted concepts $p_{neg}$. This formulation steers generation away from $c_{neg}$ and toward $c_{pos}$ by amplifying their differences.

## 3.2 VLM-GUIDED ADAPTIVE NEGATIVE PROMPTING

To generate a creative image from a given prompt $p_{pos}$, we sample initial Gaussian noise $x_T \sim \mathcal{N}(0, I)$ and initiate an augmented denoising process in which, at each denoising step, we dynamically steer the generation away from common visual concepts identified through VLM analysis, as

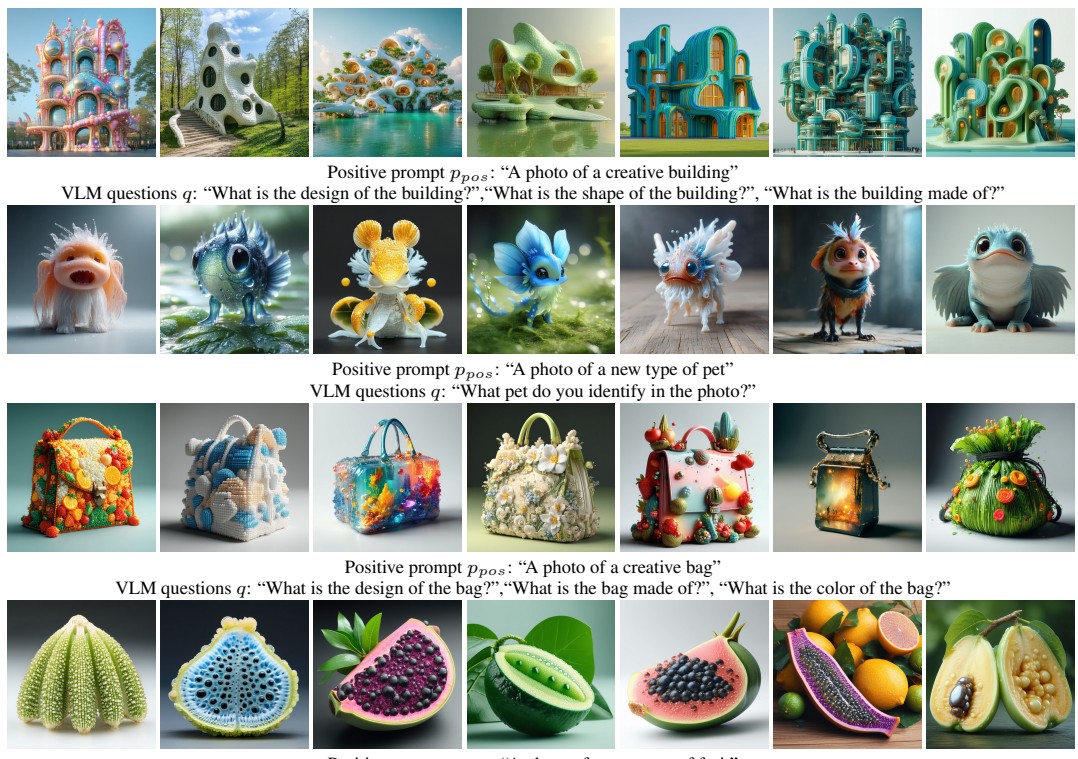

Positive prompt $p_{pos}$: "A photo of a creative building"
VLM questions $q$: "What is the design of the building?","What is the shape of the building?", "What is the building made of?"

Positive prompt $p_{pos}$: "A photo of a new type of pet"
VLM questions $q$: "What pet do you identify in the photo?"

Positive prompt $p_{pos}$: "A photo of a creative bag"
VLM questions $q$: "What is the design of the bag?","What is the bag made of?", "What is the color of the bag?"

Positive prompt $p_{pos}$: "A photo of a new type of fruit"
VLM questions $q$: "What fruit do you identify in the photo?"

Figure 4: Qualitative results of our method across different object categories. In all categories, our method generates creative shapes and appearances while preserving object semantics. For instance, buildings with unique forms and textures that retain windows, doors, and balconies, or bags made of varied materials that remain recognizable as bags.

illustrated in Figure 3. Given the intermediate prediction $\hat{x}_0^{(t)}$, at each timestep $t \in [0, T]$, we query the VLM to identify the dominant features present in the image. We denote the questioning process as follows:

$$r^{(t)} = \mathcal{V}\left(\hat{x}_0^{(t)}, q^{(t)}\right),\tag{3}$$

Where $\mathcal{V}$ is the VLM model, $q^{(t)}$ is the question, and $r^{(t)}$ is the VLM response at timestep $t$. Each response $r^{(t)}$ is added to a growing set of negative prompts: $p_{neg}^{(t)} = p_{neg}^{(t+1)} \cup r^{(t)}$ with initialization $p_{neg}^{(T)} = \varnothing$. This creates a feedback loop where each timestep's guidance reflects all previously identified dominant features, progressively steering toward more creative outputs.

**Runtime Analysis.** Our method adds minimal overhead of 13 seconds when used in the *least* efficient setting. Querying ViLT (Kim et al., 2021) for 28 steps while using the SD3.5-large decoder for $x_0$ predictions takes a total of 35 seconds, compared to 22 seconds for standard SD3.5-large single image generation. In contrast, (Richardson et al., 2024) requires approximately 8 minutes to train each concept on a single seed, and C3 requires approximately 30 minutes for amplification factor search using 10 samples per concept. A full analysis can be found in Appendix B.4.

## 4    EXPERIMENTS

We comprehensively evaluate our approach through qualitative comparisons with existing creative generation methods, a user study, and quantitative metrics. We validate our design choices with extensive ablations examining the necessity of the VLM feedback A.1, seed-specific adaptation A.2, the accumulation strategy A.3, different positive prompts A.5, robustness to different VLM models A.6, the effect of question design on the final output A.7, analysis of the VLM response on blurry predictions A.8, and analysis of the VLM querying frequency Appendices A.4 and B.1. Finally, we present use cases and practical applications that our approach enables, extending the

capabilities of previous creativity methods. Additional results and implementation details are in Section 5 and Appendices A to C.

We display in Figure 4 the diverse creative outputs of our approach across categories ranging from pets to bags. Through seed variation alone, our method explores a wide spectrum of novel concepts without requiring retraining or additional optimization.

### 4.1 QUALITATIVE EVALUATION

We begin by comparing our method with the two competing approaches for exploratory creativity within a category: ConceptLab (Richardson et al., 2024) and C3 (Han et al., 2025). As can be seen in Figure 6, ConceptLab generates creative objects but often sacrifices category validity. For example, it may produce a cup that cannot be drunk from or a couch with no seat. In contrast, our method produces objects that are both valid and creative. For fair comparison, we use the same base models as ConceptLab and C3, while also demonstrating that our method leverages newer models to produce better results. ConceptLab and C3 have several assumptions preventing them from integrating seamlessly to any base diffusion model.

In Figure 7, we compare our method with images generated by state-of-the-art models, including Stable Diffusion 3.5 (Esser et al., 2024), FLUX.1-dev (Black Forest Labs, 2024), and GPT-4o (OpenAI, 2024), all prompted with requests for "creative" or "new type of" variations. These comparisons demonstrate that even the most advanced generative models, when used with standard prompting, produce typical category exemplars – such as regular cars and fruits – rather than creative variations. In contrast, our results present novelty while maintaining validity. For example, the vehicle has wheels and a space for a driver, yet does not correspond to any existing vehicle type.

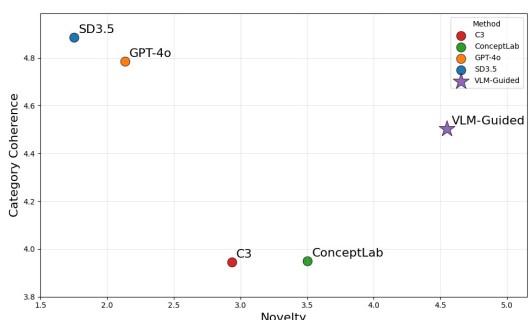

Figure 5: Trade-off between novelty and category coherence in our user study. Higher values are better for both axes. Our method (star) uniquely achieves high scores on both dimensions compared to other creative generation methods.

### 4.2 USER STUDY

Quantitative evaluation remains a fundamental challenge in computational creativity research (Lamb et al., 2018). We conduct a user study to evaluate the human-perceived creativity and semantic validity of images generated by our VLM-guided approach compared to existing methods. We collected a total of 3,200 responses (25 participants × 32 image pairs × 4 comparisons), across 8 different categories. The full setup is described in Appendix D. For each image pair, participants

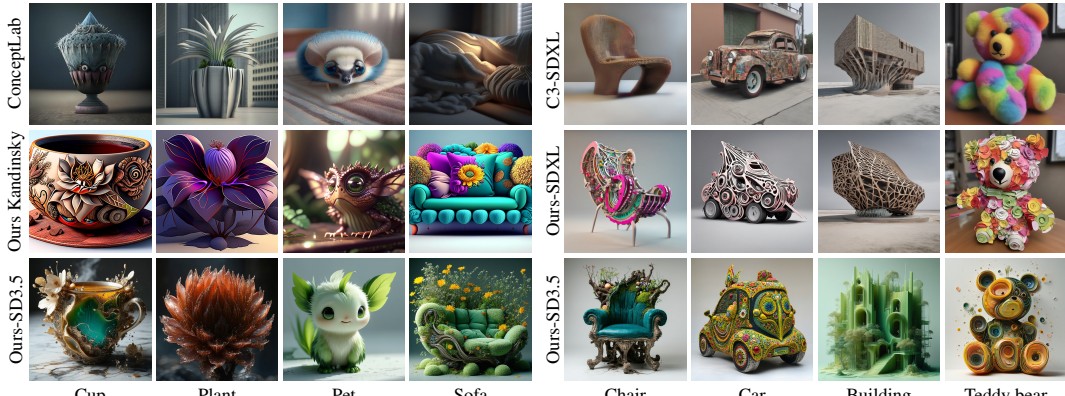

Figure 6: **Left:** Comparison with ConceptLab (Richardson et al., 2024) (top row) and our VLM-Guided method using Kandinsky2 (Razzhigaev et al., 2023) (middle row) and SD3.5 (bottom row). **Right:** Comparison with C3 (Han et al., 2025) using SDXL (Podell et al., 2023) (top row) and our method using SDXL (middle row) and SD3.5 (bottom row). Our method consistently generates more diverse and imaginative variations while maintaining recognizability within each category.

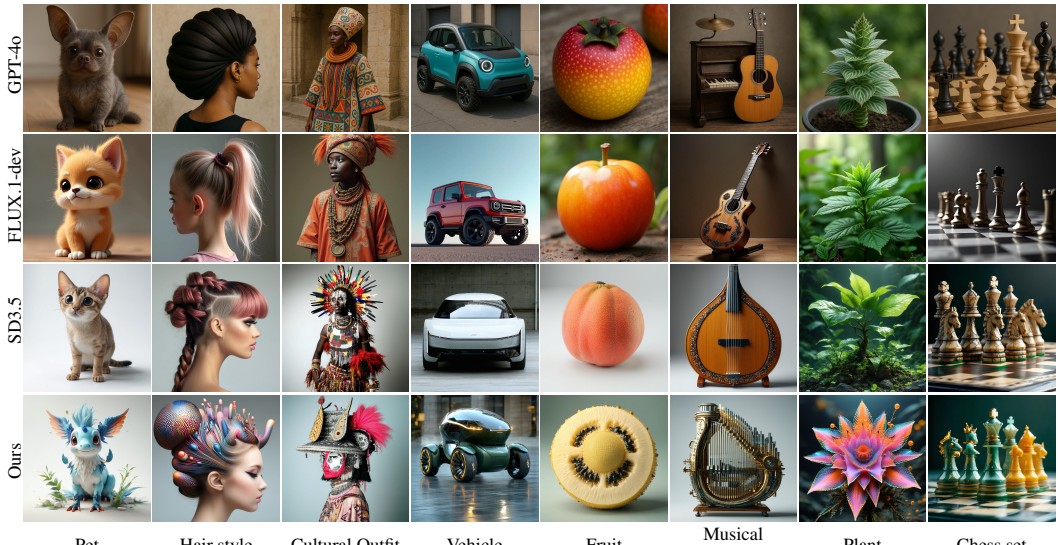

Figure 7: Creative generation comparison across different categories. Despite prompts explicitly requesting novelty ("A new type of [category]" or "A creative [category]"), GPT-4o, FLUX and SD3.5 produce typical category exemplars. Our method generates novel variations that navigate unexplored modes of the semantic space. Each column uses identical seeds across all methods for fair comparison.

evaluate Creativity/Novelty: How creative or novel is the interpretation of the broad category? and validity: How well does the image maintain its identification as the specified category? Figure 5 presents the results. "Creative Prompting" methods (SD3.5 and GPT-4o), explicitly requesting novelty via prompts such as "A new photo of a [category]", cluster in the upper-left region with high category validity but minimal novelty, confirming our qualitative findings that simple prompt modifications fail to produce creative exemplars. Creative-generation methods (ConceptLab and C3) achieve moderate creativity results but at a significant cost in validity. In contrast, our method achieves both high novelty and validity, maintaining both high creativity and validity.

### 4.3 ABLATION STUDIES

A natural question is whether the in-the-loop VLM guidance is necessary or does one of two offline alternatives suffices: (i) using an LLM to derive a negative list from the positive prompt alone, or (ii) using a VLM to analyze a random image once and then statically replaying the resulting list across all seeds. We study four design variants to validate our adaptive negative prompting approach, as presented in Figure 8. First, we tested whether GPT-4o could generate static negative prompt lists directly from the main object in the positive prompts. Second, applying our accumulated negative prompts statically (replaying) from the beginning yields less creative outputs. Third, reusing negative prompts across different seeds (Cross-Seed replay) produces suboptimal results. Finally, removing accumulation allows generations to cycle back to the conventional patterns previously identified. Our method achieves the best scores across all reported metrics in Table 1.

The full ablation studies are presented in Appendix A. They examine computational efficiency (i.e., timestep reduction), VLM robustness across different models, question design impact, and positive prompt variations, all confirming the robustness of our approach.

### 4.4 QUANTITATIVE EVALUATION

Existing methods employ different strategies to quantify and evaluate creativity. ConceptLab measures the difference between CLIP similarity to the positive concept prompt and the maximum CLIP similarity to any negative concept prompt. We refer to this measure as "relative

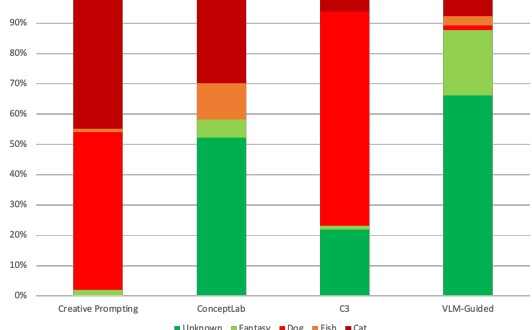

Figure 9: Top 5 subcategory distribution of 100 generated pets per method classified with GPT-4o.

Table 1: Quantitative evaluation of creative generation methods across different prompting strategies. Reference: SD3.5 with "A photo of a [category]". Creative Prompting: SD3.5 with "A photo of a creative [category]". VLM-Guided: Our adaptive negative prompting approach. C3 and ConceptLab images are generated as explained in the corresponding papers. The metrics are averaged over 400 samples, equally generated 100 from 4 categories: pet, plant, garment, vehicle. In **bold** are best results underline for second best, within each base-model category. For validity we exclude the baselines (Reference & Creative Prompting) from the marking.

| Method | Novelty | | Diversity | | Validity | |
|---|---|---|---|---|---|---|
| | Relative Typicality ↑ | GPT Novelty Score ↑ | Total Variance ↑ | Vendi ↑ | CLIP Score ↑ | GPT Score ↑ |
| ConceptLab-Kandinsky2 | 1.922 | 0.238 | 0.289 | 5.119 | 0.270 | 0.862 |
| **Stable Diffusion 3.5 Large Base Model** | | | | | | |
| Reference SD3.5 | 1.640 | 0.065 | 0.188 | 3.174 | 0.282 | 1.000 |
| Creative Prompting SD3.5 | 1.645 | 0.230 | 0.191 | 3.139 | 0.267 | 0.933 |
| GPT-4o 10 Concepts | 0.655 | 0.093 | 0.272 | 4.973 | 0.262 | 0.867 |
| GPT-4o 15 Concepts | 0.885 | 0.108 | 0.277 | 5.040 | 0.262 | 0.805 |
| GPT-4o 28 Concepts | 1.043 | 0.100 | 0.276 | 5.067 | 0.260 | 0.828 |
| Cross-Seed Replay | 1.703 | 0.065 | 0.265 | 4.584 | 0.261 | 0.843 |
| No Accumulation | 1.610 | 0.060 | 0.274 | 4.355 | 0.262 | 0.875 |
| Captions Regeneration | 1.317 | 0.187 | 0.279 | 5.020 | 0.248 | 0.663 |
| Ours SD3.5 + ViLT | 1.835 | 0.157 | 0.298 | 5.347 | **0.264** | 0.893 |
| Ours SD3.5 + BLIP-1 | 2.005 | 0.230 | 0.299 | 5.414 | **0.264** | 0.856 |
| Ours SD3.5 + BLIP-2 | **2.190** | 0.370 | **0.318** | **5.794** | 0.261 | 0.898 |
| Ours SD3.5 + Qwen2.5 | 2.100 | **0.401** | 0.308 | 5.476 | **0.264** | **0.917** |
| **SDXL-1.0 Base Model** | | | | | | |
| Reference SDXL | 1.775 | 0.015 | 0.174 | 2.906 | 0.283 | 1.000 |
| Creative Prompting SDXL | 1.540 | 0.155 | 0.206 | 3.640 | 0.274 | 0.9125 |
| C3-SDXL | 1.075 | 0.232 | 0.271 | 4.726 | **0.254** | **0.895** |
| Ours SDXL + Qwen2.5 | **1.795** | **0.405** | **0.296** | **5.427** | 0.252 | **0.895** |

typicality". C3 evaluates three dimensions of creativity: novelty, diversity, and validity. We evaluate creativity through complementary metrics that capture novelty, diversity, and validity as well. For novelty, we measure relative typicality (multiplied by 100 for readability) and the GPT Novelty Score. For the diversity we measure Vendi score and total variance. For validity, we employ CLIP alignment and GPT-4 verification. While these metrics have known limitations for creative outputs, as creativity inherently deviates from training distributions, they provide consistent comparative baselines. The formal definitions of the metrics are presented in Appendix E.

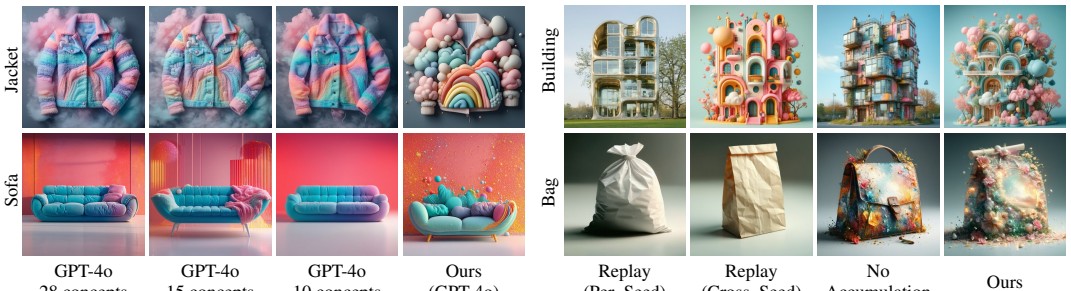

Figure 8: **Left:** Non-Adaptive LLM Approach: GPT-4o ($n \in [10, 15, 28]$) - static LLM list of $n$ negative concepts applied at all steps. Ours (GPT-4o) dynamic, VLM-guided negatives using GPT-4o as our VLM. **Right:** Replay (Per-Seed) - reuse the accumulated VLM list from the *same* seed at all steps; Replay (Cross-Seed) - reuse a list extracted from a *different* seed at all steps; No Accumulation - use only the current step's VLM answers (no carry-over); Ours - adaptive accumulation of negative prompts.

**Quantitative Results**    Table 1 summarizes the quantitative results. We achieve significant improvements in diversity and novelty metrics with minimal tradeoff in CLIP and GPT scores. All metrics are averaged across four categories: "vehicle", "plant", "pet", and "garment" (100 images each), so improvements reflect cross-category behavior. We divide our results based on the backbone base model used for the generation. When applied to SDXL (Podell et al., 2023) our method still achieves significant improvements, even though SDXL is an inherently less capable and less diverse base model than SD3.5, our method still produces improvements. This variant also outperforms C3-SDXL across novelty and diversity metrics while maintaining comparable validity scores. This demonstrates that our method promotes creative exploration regardless of the base model. Our method using SD3.5+Qwen2.5-3B and SD3.5+BLIP-2 achieves the best balance across all three creativity dimensions, leading in novelty and diversity, and maintaining competitive validity, while other methods either sacrifice creativity for validity or vise versa. The design variants we evaluate under-perform our dynamic, per-step, per-seed approach, highlighting the importance of both timing and seed-specific guidance. A no-accumulation variant also trails our method, indicating that remembering previously discovered negatives is beneficial. Notably, while ConceptLab achieves the highest CLIP score, it shows the lowest GPT verification score. This happens because their optimization process maximizes the CLIP-space distance from negative concepts but can produce adversarial examples that satisfy mathematical constraints without maintaining semantic validity. This manifests as objects that technically align with CLIP embeddings but fail human and GPT-4 verification as functional category members (e.g., cups without cavities and sofas without seating surfaces). In contrast, our method maintains the highest performance across all three evaluation dimensions: "validity", "diversity", and "novelty". The caption regeneration experiment demonstrates a limitation of pre-determined prompting: we used Qwen2.5-VL (Bai et al., 2025) to generate detailed captions of our creative images, then attempted to regenerate those images from the captions alone. Despite having detailed descriptions of creative objects, explicitly prompting for creativity, the regenerated images show substantially lower novelty and validity scores. This demonstrates that even very detailed text prompts cannot replicate the creative exploration achieved by our adaptive guidance approach.

**GPT Novelty Score**    In Figure 9, we present the distribution of subcategories classified with GPT-4o over 100 images of pets generated with ConceptLab, C3, Creative Prompting, and Our VLM-Guided method. While Creative Prompting and C3 generate recognizable dogs and cats, with ConceptLab exhibiting intermediate behavior, our approach primarily produces unknown or unclassifiable pets, approximately 87%, demonstrating our method's ability to avoid known subcategories.

## 4.5    Use Cases

**Diverse scenarios.**    Our method generates novel objects within semantic categories and can be used for practical applications by placing these objects in diverse contexts and scenes. Recent controllable generation models like Flux.1-dev Kontext (Black Forest Labs, 2025) enable users to take our creatively generated objects and seamlessly integrate them into various environments while preserving their unique characteristics. Interestingly, we found that this approach achieves better consistency compared to ConceptLab's method of integrating optimized tokens into different prompts. We show an example of this phenomenon in Figure 10. Each building that is generated by reusing the textual token is different than the other. On the other hand, using Flux-Kontext our creative building looks consistent throughout the scenes.

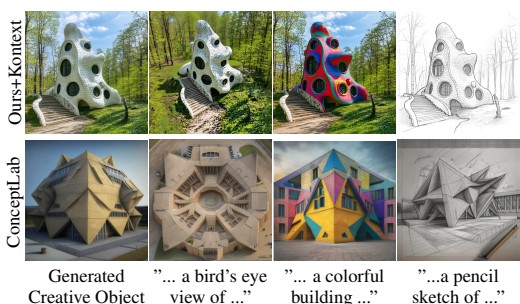

Figure 10: Creative object in different scenes. Left column: Novel objects generated by our VLM-guided method and reused with Flux-Kontext (Black Forest Labs, 2025) (top row) and ConceptLab (bottom row).

**Complex prompts.**    Figure 11 displays how our VLM-guided approach seamlessly integrates with elaborate prompt descriptions, "A photo of an imaginary pet surfing on a board near an island", "A photo of a new type of plant

Figure 11: Creative objects presented in a complex environment described by the prompt.

blooming in an arctic field next to penguins", "A photo of a woman wearing a creative jacket in a french cafè" and "A photo of a new type of fruit sliced on a ceramic plate", enabling creative exploration even within complex requirements.

The adaptive negative prompting mechanism operates orthogonally to these additional constraints, it identifies and steers away from conventional modes of the requested object described as "creative", while respecting the stylistic and compositional requirements specified in the prompt.

To evaluate our method's controllability, we constructed a benchmark of 200 diverse complex prompts spanning categories such as animals, plants, fashion, and food, each embedding creative elements within elaborate scene descriptions (e.g., "A photo of a creative insect

| Method | VIE-SC | VIE-PQ | Total |
|---|---|---|---|
| Creative Prompting | 8.992 | **8.659** | 8.769 |
| Ours (SD3.5+ViLT) | **9.163** | 8.609 | **8.848** |

Table 2: VIE scores on 200 complex prompts.

resting on a dew-covered leaf in a quiet morning meadow"). We evaluate prompt adherence and perceptual quality using VIEscore (Visual Instruction-guided Explainable) scores (Ku et al., 2024). As shown in Table 2, our method achieves higher VIE-SC scores compared to creative prompting with SD3.5 alone, while maintaining comparable perceptual quality (VIE-PQ). This demonstrates that our adaptive negative prompting generates central objects while respecting complex scene descriptions and compositional constraints which aligns better with the prompt requesting for creativity. Full benchmark construction details and automated question generation methodology are provided in Appendix E.6.

**Beyond single objects.** Our method extends naturally from generating individual creative objects to producing coherent sets of related items that share a unified creative vision. By applying our approach to prompts that describe collections e.g., "Creative tea set", as presented in Figure 12, we demonstrate that our method maintains validity and consistency across multiple objects while exploring creative variations.

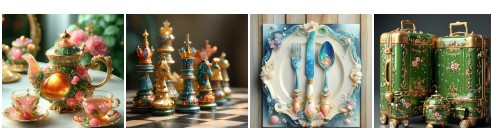

| Tea set | Chess set | Cutlery set | Luggage set |

Figure 12: Creative sets generated by our method demonstrating coherent collections of related objects. Each set exhibits individual creativity in its components while maintaining stylistic and functional consistency across the collection.

## 5 CONCLUSIONS

We introduced VLM-Guided Adaptive Negative-Prompting, an inference-time method that leverages the strength of vision-language models to dynamically steer diffusion models toward more creative outcomes. By querying a VLM throughout the denoising process and accumulating seed-specific negative prompts, our approach pushes generation away from conventional patterns while preserving categorical coherence. The fact that a VLM is capable of analyzing noisy intermediate states and providing guidance strong enough to redirect the trajectory highlights its potential as a powerful mechanism for creative exploration.

While our VLM-guided approach demonstrates effective creative exploration, several limitations can be addressed in future research. First, our method introduces computational overhead through VLM inference at each timestep, though our ablation studies show this can be reduced to the first 10-15 steps without significant quality loss. Second, the quality of creative outputs depends on the VLM's ability to identify emerging patterns in noisy intermediate predictions; while we demonstrate robustness across various VLMs, more sophisticated vision-language models generally yield better results. Third, our approach requires careful question design for optimal performance; different question formulations work better for different semantic categories, and automating this selection remains an open challenge.

Looking ahead, we believe that the integration of feedback-driven guidance will open new directions for creativity in generative models, and future work may extend this paradigm to other domains, such as video, 3D, or multimodal content creation.

**Reproducibility Statement**    We provide the necessary details to reproduce our results. Algorithmic steps and Hyperparameters (feedback window, frequency $f$, accumulation, replay variants) are specified in the main text and Appendix B; evaluation protocols, metrics, and prompts are in Appendix C. We release per-category negative lists (static and accumulated) and the exact question templates used by the VLM in Appendix A. Random seeds, category splits, and generation counts are stated in the implementation details (Appendix B). We will release the code of our project in the near future.

**Ethics Statement**    Our study involves image generation within broad, non-sensitive categories. We avoid instructions and outputs that target protected attributes or hazardous content. The user study (Appendix D) followed institutional guidelines: no personally identifying information was collected, and data were anonymized and aggregated for analysis. All third-party models and datasets were used under their respective licenses, and we disclose model choices and prompts (Appendices B and C). We report compute and runtime to enable the assessment of environmental impact (Appendix B). No conflicts of interest or external sponsorship influenced the findings.

## ACKNOWLEDGMENTS

We thank Roy Ganz, Guy Ohayon, Omer Belhasim and Tomer Borreda for their early feedback and helpful suggestions. This work was supported by a research gift from Adobe.

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

# APPENDIX

This appendix provides comprehensive details supporting our main paper. Section A presents extensive ablation studies. Section B provides technical implementation specifications. Section C details the qualitative evaluation framework and the generation process of the evaluated methods. Section E details evaluation metrics. Section D describes our human evaluation protocol.

## A ABLATIONS

### A.1 NON-ADAPTIVE LLM APPROACH.

Table 3: Exact GPT–4o lists used as $p_{\text{neg}}^{\text{LLM}}$ for the Jacket category in Figure 8.

| $N$=10 | $N$=15 | $N$=28 |
|---|---|---|
| bomber, biker, trucker, parka, puffer, blazer, varsity, trench, anorak, field | bomber, biker, trucker, parka, puffer, blazer, varsity, trench, anorak, field, harrington, peacoat, safari, quilted, windbreaker | bomber, biker, trucker, parka, puffer, blazer, varsity, trench, anorak, field, harrington, peacoat, safari, quilted, windbreaker, denim, leather, fleece, rain, down, coach, double breasted, chore, utility, cagoule, car, duffle, mac |

Table 4: Exact GPT–4o lists used as $p_{\text{neg}}^{\text{LLM}}$ for the Sofa category in Figure 8.

| $N$=10 | $N$=15 | $N$=28 |
|---|---|---|
| sectional, loveseat, chaise, recliner, futon, sleeper, modular, tuxedo, chesterfield, camelback | sectional, loveseat, chaise, recliner, futon, sleeper, modular, tuxedo, chesterfield, camelback, lawson, midcentury, slipcovered, daybed, settee | sectional, loveseat, chaise, recliner, futon, sleeper, modular, tuxedo, chesterfield, camelback, lawson, midcentury, slipcovered, daybed, settee, track arm, roll arm, armless, curved, divan, sofa bed, pit, pallet, reclining, convertible, chaise end, bench, ottoman |

We used GPT-4o (OpenAI, 2024) to generate lists of common sub-categories for each creative prompt at several sizes $N \in [10, 15, 28]$. For instance, given the prompt "A photo of a creative jacket", we asked GPT-4o: "List the $N$ most common types of jackets. A single list, separated by commas. Each description is a single word". A typical result is: "bomber, biker, trucker ...". We then formatted the list as a static negative prompt $p_{neg}^{LLM}$ and applied it uniformly throughout the entire denoising process $p_{neg}^{(0)} = p_{neg}^{(1)} = \cdots = p_{neg}^{(T)} = p_{neg}^{LLM}$. As shown in Figure 8, this approach produces less creative results compared to our dynamic method. For example, in the second row, our generated jacket features smooth, cloud-like spherical ornaments that are atypical for jackets, whereas LLM-based lists yield colorful yet conventional wool or fabric designs and do not portray creative ornaments. We attribute this to the lack of alignment between the static, seed-independent LLM-generated list and the actual generative trajectory. Such prompts cannot account for the specific visual patterns that emerge during the denoising process, nor for those encoded in the sampled

initial noise. While the LLM provides semantically reasonable negative concepts, it lacks the visual awareness to recognize which particular modes are being generated from the specific sampled noise at each timestep, resulting in generic rather than targeted steering.

## A.2 NON-DYNAMIC REPLAY APPROACHES.

To isolate the importance of the dynamic process, we tested whether the accumulated negative prompts from our full dynamic negatives list could be replayed statically from the beginning of the generation. In this experiment, we first ran our complete dynamic method to generate the final accumulated negative prompt $p_{neg}^T = \bigcup_{t=1}^{T} p_{neg}^{(t)}$ for a given seed, then used this pre-accumulated prompt uniformly throughout a fresh denoising process: $p_{neg}^{(t)} = p_{neg}^{(T)}$ for all timesteps $t \in [0, T]$. Despite using the same negative concepts that our dynamic method accumulates, this static application produces less creative results. For example, the bag in Figure 8 in the last row generated with the adaptive method has flower ornaments and a unique shape while the bag under the "Replay (Per-Seed)" column looks like a regular plastic bag. This demonstrates that timing and responsiveness to emerging visual patterns are crucial; the same negative prompts, when applied at the wrong times, fail to provide effective steering. The dynamic nature of our approach, which introduces negative concepts precisely when the corresponding visual patterns begin to emerge, is essential for successful creative exploration. We further investigate whether negative prompts can be reused across different generation seeds to reduce computational overhead. We collected accumulated negative prompts $p_{neg}^{(T)}$ from successful creative generations and applied them to random seeds while maintaining the same positive prompt. This cross-seed reuse consistently produces suboptimal results, emphasizing that each generation seed follows a unique trajectory through the semantic space and requires its own adaptive negative prompting strategy. When the VLM's analysis of intermediate predictions $\hat{x}_0^{(t)}$ is tailored to the specific seed's denoising path, we achieve superior creative results, as shown in Figure 8 under the column "Replay (Cross-Seed)". For example, the bag in the last row under the "Replay (Cross-Seed)" column looks like a regular paper bag compared to our unique bag design. This finding reinforces the notion that the effectiveness of our method stems from its ability to provide adaptive, trajectory-specific guidance rather than applying generic steering patterns.

## A.3 NON-ACCUMULATING APPROACH.

Next, we explore the importance of our accumulation strategy. To test its contribution, we modify our approach to use only the current VLM response as the negative prompt at each timestep. Specifically, we replace the negative prompt with $p_{neg}^{(t)} = r^{(t)}$ for each $t \in [0, T]$, discarding all previously accumulated information. This non-accumulating variant, shown in Figure 8 under the column "No Accumulation", fails to maintain a memory of previously identified conventional modes, allowing the generation to cycle back toward familiar patterns that were detected and should have been avoided in earlier denoising steps. For example, the building in the first row under the column "No Accumulation" remains similar to the SD3.5 baseline building, whereas our method produces a

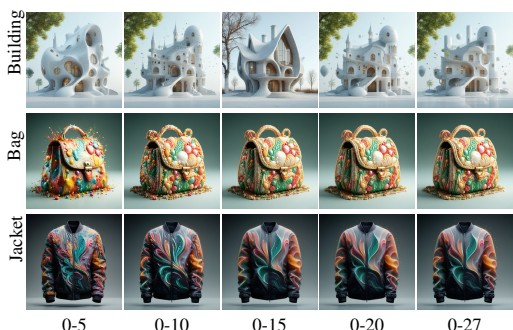

Figure 13: Effect of limiting VLM guidance to different ranges of denoising timesteps. Columns correspond to applying our method during only the first 5, 10, 15, 20, or all 27 timesteps, while rows show results for Building, Bag, and Jacket categories.

unique, asymmetrically shaped building. For a fair comparison, the VLM query is identical across methods: at every timestep, we ask "What type of bag is this?".

## A.4 TIMESTEPS ANALYSIS.

Our method introduces VLM evaluations at each denoising timestep, which unavoidably increases computational overhead compared to standard diffusion sampling. To improve practical efficiency, we investigate whether the number of VLM queries can be reduced without compromising creative quality.

Table 5: Exact GPT-4o lists used as $p_{\text{neg}}^{\text{LLM}}$ for all categories in the LLM ablation study presented in Table 1.

| Category | N=10 | N=15 | N=28 |
|---|---|---|---|
| Pet | dog, cat, fish, bird, rabbit, hamster, guinea pig, turtle, lizard, snake | dog, cat, fish, bird, rabbit, hamster, guinea pig, turtle, lizard, snake, parrot, ferret, chinchilla, hedgehog, tarantula | dog, cat, fish, bird, rabbit, hamster, guinea pig, turtle, lizard, snake, parrot, ferret, chinchilla, hedgehog, tarantula, gecko, bearded dragon, cockatiel, budgerigar, finch, tortoise, newt, axolotl, hermit crab, dwarf hamster, betta, goldfish, lovebird |
| Plant | tree, shrub, grass, fern, moss, cactus, succulent, vine, herb, flower | tree, shrub, grass, fern, moss, cactus, succulent, vine, herb, flower, palm, orchid, bamboo, lily, rose | tree, shrub, grass, fern, moss, cactus, succulent, vine, herb, flower, palm, orchid, bamboo, lily, rose, tulip, daisy, sunflower, maple, oak, pine, conifer, broadleaf, evergreen, deciduous, ivy, sedge, reed |
| Garment | shirt, dress, pants, skirt, jacket, coat, sweater, hoodie, t-shirt, blouse | shirt, dress, pants, skirt, jacket, coat, sweater, hoodie, t-shirt, blouse, jeans, shorts, suit, cardigan, jumpsuit | shirt, dress, pants, skirt, jacket, coat, sweater, hoodie, t-shirt, blouse, jeans, shorts, suit, cardigan, jumpsuit, blazer, trenchcoat, parka, raincoat, overcoat, waistcoat, sweatshirt, tracksuit, leggings, chinos, dungarees, kimono, sari |
| Vehicle | car, truck, bus, van, motorcycle, bicycle, scooter, train, tram, subway | car, truck, bus, van, motorcycle, bicycle, scooter, train, tram, subway, boat, ship, ferry, airplane, helicopter | car, truck, bus, van, motorcycle, bicycle, scooter, train, tram, subway, boat, ship, ferry, airplane, helicopter, yacht, canoe, kayak, jet, glider, seaplane, submarine, hovercraft, snowmobile, atv, forklift, tractor, bulldozer |

Table 6: Accumulated lists reused for static application in Fig. 8.

| Category | Accumulated negative list |
|---|---|
| Building | `brick, regular building, glass, modern, skyscraper, concrete, moderne, modernist, futuristic, curved` |
| Bag | `tote, satchel, hobo, backpack, clutch, messenger, crossbody, duffel, bucket, wristlet` |

Specifically, we analyze the minimum number of timesteps requiring VLM intervention to achieve effective creative steering. As shown in Figure 13, applying VLM guidance during only the first 10 to 15 timesteps sufficiently steers generation toward creative outputs. This efficiency results from the momentum effect described in (Ban et al., 2024) and explained in our Section 2, where early negative prompt accumulation establishes persistent creative trajectories that continue throughout the remaining denoising process. This finding enables improved computational efficiency, making our approach more practical for real-world deployment. For all methods in this analysis, the VLM query is identical and fixed at every queried step: "What is the style of the [category]?".

## A.5 POSITIVE PROMPT SELECTION.

Our approach demonstrates flexibility in positive prompt formulation, accepting various creativity-indicating phrases such as "creative", "innovative", "new", "novel", "unique", and other similar terms to produce creative outputs. Our VLM-guided approach works effectively even with ambiguous positive prompts, such as "a new type of...". As demonstrated in Figure 14, different formulations of creative prompts yield diverse creative outputs while maintaining the fundamental steering behavior and the effectiveness of our method as well as validity. When the indicative adjective is removed entirely from the positive prompt (e.g., using simply "A photo of [obj]"), the resulting images are diverse and aesthetically pleasing; however, they lack the creative qualities that distinguish our method.

## A.6 ROBUSTNESS TO VLM MODEL SELECTION.

Our method demonstrates robustness across a variety of Vision-Language Models that differ

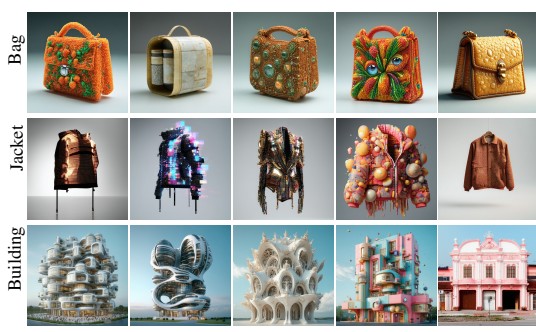

New type    Innovative    Unique    Creative    None

Figure 14: Effect of positive prompt wording on creative generation using our method. Columns correspond to alternative prompt formulations ("New type", "Innovative", "Unique", "Creative" and simply "A photo of a [category]"), while rows show results for different semantic categories Across categories, our approach produces diverse and imaginative outputs.

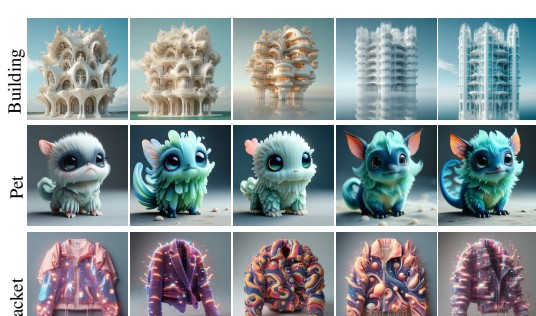

GPT-4o    Qwen2.5    BLIP-2    BLIP-1    ViLT

Figure 15: Comparison of outputs when guiding our method with different Vision-Language Models (VLMs). Columns correspond to GPT-4o (OpenAI, 2024), Qwen2.5 (Bai et al., 2025), BLIP-2 (Li et al., 2023), BLIP-1 (Li et al., 2022), and ViLT (Kim et al., 2021), while rows show three semantic categories: Unique Building, New Pet, and Creative Jacket. Across models, our approach consistently produces creative and coherent results, with stronger VLMs generally yielding more novelty, demonstrating robustness of the method to the choice of VLM.

in architecture, training data, model size, and capabilities. As shown in Figure 15, we successfully achieve creative outputs using models ranging from lightweight options such as ViLT (Kim et al., 2021) and BLIP-1 (Li et al., 2022) to more sophisticated models like BLIP-2 (Li et al., 2023), Qwen2.5 (Bai et al., 2025), and GPT-4o (OpenAI, 2024). While more capable VLMs generally produce higher quality creative results, the consistent creative steering behavior across different model choices validates the generalization capabilities of our approach. This robustness ensures that practitioners can select VLMs based on their specific computational constraints and quality

requirements while maintaining the fundamental creative exploration functionality. For all methods in this analysis, the VLM query is identical and fixed at every queried step: "What type of [category] is this?".

### A.7 QUESTION DESIGN FOR CREATIVE EXPLORATION.

The choice of question formulation is a critical design parameter that determines which visual features are identified and which are steered away from, directly influencing the creative output. Based on our empirical findings, we recommend object-focused questions (e.g., "What is the main object in this image?") for generating "new types" of variations within familiar categories(animals, furniture, buildings, etc.). Style or attribute focused questions (e.g., "What is the style/design/texture/material in this image?") are optimal for aesthetic novelty and creativity while preserving category coherence. Figure 16 presents the variations of the question $q^{(t)}$ choice and the direct influence on the output. For example, when the VLM is prompted about materials, the bag output transforms from regular leather to a knitted, colorful material.

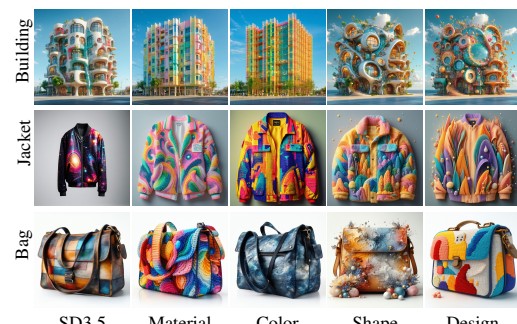

Figure 16: Effect of the VLM question design on creative generation. Rows correspond to three semantic categories. The first column shows a Stable Diffusion 3.5 baseline. The remaining columns apply our VLM-Guided Adaptive Negative-Prompting while asking the VLM about (i) the material, (ii) the dominant colors, (iii) the object's shape, and (iv) its design.

### A.8 VLM PREDICTION ANALYSIS.

To understand how our VLM-guided approach effectively steers generation despite operating on noisy intermediate predictions, we analyze the VLM's ability to identify emerging semantic patterns throughout the denoising process. We examine the correlation between VLM predictions on early, blurry $\hat{x}_0$ estimates and the final generated content across timesteps 0 to 27. Figure 17 shows that VLM correlation rapidly increases during the initial denoising steps, reaching approximately $90\%$ within the first 3 to 5 timesteps, despite the highly noisy nature of the early predictions. The high correlation between early VLM predictions and final outputs validates our approach of accumulating negative prompts from the beginning of the denoising process, as the predictions of the VLM are meaningful even under noisy conditions.

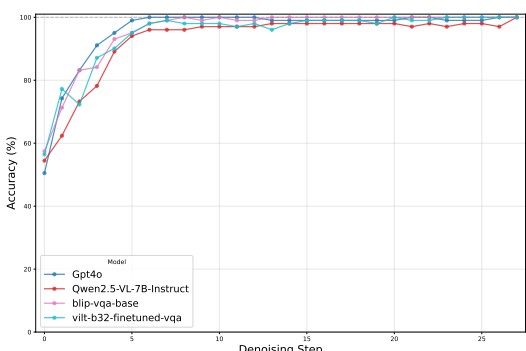

Figure 17: Correlation between the VLM answers across different timesteps and the final generated image

### A.9 CREATIVE CAPTION GENERATION

To investigate whether detailed text descriptions that prompt for creativity can reproduce our results, we conducted the following experiment: we used Qwen2.5-VL (Bai et al., 2025) to generate captions for each image we used in our main quantitative experiment (Table 1), i.e., 400 images in total. The prompt to Qwen2.5 we used is "Give a detailed caption to the image". We then used SD3.5 to generate new images from these detailed captions. This experiment evaluates empirically whether a sufficiently detailed human-written (or VLM-written) prompt can achieve the same creative results as our adaptive negative prompting approach. As shown in Table 1, the results of the captions regeneration are less diverse, novel and decrease in validity.

## B    IMPLEMENTATION DETAILS

Unless noted, experiments use SD3.5 large, 28 steps and classifier-free guidance (CFG) 4.5. The default VLM is Qwen2.5-VL-3B-Instruct; we also support BLIP2 (Li et al., 2023), BLIP1 (Li et al., 2022), ViLT (Kim et al., 2021), and GPT-4o (OpenAI, 2024). We run on a single NVIDIA A40, at $1024 \times 1024$ resolution.

### B.1    VLM FEEDBACK WINDOW.

We allow the user to query the VLM over a predefined window of steps to minimize overhead. Let $t_{\text{start}}$ and $t_{\text{stop}}$ be the step indices when both are provided; otherwise, they are set by default to 0 and 28. Within this window we query at a fixed frequency $f$. The default is set to $f = 1$ (every step), but users may increase $f$ to reduce calls (e.g., every 2 or 4 steps). The feedback window and frequency integrate directly into our guidance loop; see 3 for how VLM answers are accumulated and applied.

### B.2    ADAPTIVE NEGATIVE PROMPTING CONSTRUCTION.

At each step $t \in [0, T]$, we decode $\hat{x}_0$ to RGB and ask a set of questions $\{q_i\}^{(t)}$. We then apply a light normalizer: remove unwanted prefixes, e.g., "it looks like", drop leading articles, and collapse whitespace and punctuation. We maintain a single negative prompt string, containing a list of $\mathcal{N}$ negatives with: (i) case-insensitive deduplication, (ii) re-encoding only when $\mathcal{N}$ changes, and (iii) all the negatives are separated by commas. During the VLM feedback window, we update the negative half of the CFG embedding pair from the comma-joined string of $\mathcal{N}$ negatives and keep the positive half unchanged. When leaving the VLM feedback window, we clear the negative prompt and replace it with an empty string.

### B.3    DECODING $\hat{x}_0$: VAE VS. LINEAR APPROXIMATION.

The diffusion model operates in latent space. Therefore, obtaining clean image predictions $\hat{x}_0$ for input to the VLM requires passing them through the VAE decoder, which is costly at every denoising step. Prior works (Vass, 2024; Turner, 2022) have empirically shown that the decoders of common text-to-image diffusion models can be well-approximated by a linear transformation, enabling significant acceleration of the decoding process. For example, Vass (2024) showed that, in the case of SDXL, this linear transformation can be expressed by the matrix:

$$w = \begin{bmatrix} 60 & -60 & 25 & -70 \\ 60 & -5 & 15 & -50 \\ 60 & 10 & -5 & -35 \end{bmatrix}.$$

A similar linear transformation can be applied to SD3.5 with a different weight matrix. In our method, using this linear approximation yields creative results comparable to those obtained with the full decoder, while substantially reducing computational overhead.

### B.4    FULL RUNTIME ANALYSIS.

Our method adds only modest overhead in the lightweight-VLM regimes (ViLT/BLIP-1/BLIP-2), and reducing the amount of querying offers a simple, effective way to trade compute for guidance strength.

### B.5    VLM-QUERYING AUTOMATION

To make our method as easy to use as standard text-conditioned diffusion generation, we added the option for automated question generation. The user passes a creative prompt (e.g., "A photo of a creative animal") as an argument to the model, and an LLM (GPT-4o (OpenAI, 2024)) automatically generates VLM queries without manual tuning. The pipeline consists of three steps: first, we extract the main object we aim to focus on from the positive prompt, i.e., we will extract "[animal]" in this example. Then, we provide the LLM with the analysis we conducted on the question design as context. The LLM is requested to generate similar questions appropriate for the specific main object. We then pass those questions as an argument to our creative-generation pipeline.

Table 7: Runtime with VLM-in-the-loop guidance. Total seconds for SD3.5-large single-image generation when querying different VLM oracles at either every denoising step (28) or only the early steps (15). The baseline performs no VLM queries. All runs use the same prompt and seed.

| VLM | Steps | Runtime (Seconds) |
|---|---|---|
| Baseline No VLM | 28 | 22 |
| ViLT | 28 | 35 |
| | 15 | 29 |
| BLIP-1 | 28 | 36 |
| | 15 | 30 |
| BLIP-2 | 28 | 43 |
| | 15 | 33 |
| Qwen2.5-3B | 28 | 71 |
| | 15 | 56 |

### B.6 COMPLEX-PROMPTS BENCHMARK

To quantitatively evaluate controllability in complex scenarios, we created a benchmark of 200 diverse prompts that test whether our method can maintain prompt adherence while modifying only the central objects to be creative. To construct such benchmark we prompted GPT-4o to provide with 200 diverse object categories, similar to those present in the paper (for example, animals, hairstyles, accessories, etc.). Each prompt follows the template "A photo of a [creative/innovative/new type of/novel/unique] [main object] [scene description]". The template was filled by GPT-4o, according to the query "Write a prompt using the template: A photo of a [creative/innovative/new type of/novel/unique] [main object] [scene description]. Choose an appropriate creativity indicator from the list [creative/innovative/new type of/novel/unique], and place the object in logically feasible scene. Describe it briefly." For each prompt we generated an automatic questions list using the method described in the previous section.

Example Prompts: "A photo of an imaginary pet resting inside a terrarium filled with miniature plants.", "A photo of a creative hairstyle showcased on a model standing in a sunlit desert landscape." "A photo of a creative insect resting on a dew-covered leaf in a quiet morning meadow."

## C QUALITATIVE EVALUATION FRAMEWORK

For a fair evaluation, we adopt each baseline's evaluation setting including their prompts, models and experimental protocols. Specifically, we use their original prompts: "a creative [obj]" for C3 and "Professional high quality photo of a new type of [obj]. photorealistic, HQ, 4k" for ConceptLab. We also integrate our method into their respective models: SDXL (Podell et al., 2023) for C3 and Kandinsky 2.1 (Razzhigaev et al., 2023) for ConceptLab. We note that ConceptLab's method leverages Kandinsky's Diffusion Prior model, which their optimization process specifically requires for learning creative concepts in the prior's output space (Richardson et al., 2024). To ensure direct comparability, we integrate our method into each baseline's model and generate samples using identical seeds. Additionally, we showcase our method's full potential using Stable Diffusion 3.5 (Esser et al., 2024), demonstrating superior creative generation with state-of-the-art architectures.

## D USER STUDY

Participants view pairwise comparisons of images generated from the same broad category (e.g., "pet", "building", "vehicle"). Each comparison shows outputs from our method versus one of the four baselines. Creative prompts: SD3.5 and GPT-4o using "A photo of a creative/new type of [category]" and creative generation methods: ConceptLab and C3.

Table 8: User study results showing average ratings (1-5 scale) for novelty and category coherence. Our method achieves the highest novelty while maintaining strong categorical identity.

| Method | Novelty ↑ | validity ↑ |
|---|---|---|
| SD3.5 | 1.753 | 4.886 |
| GPT-4o | 2.133 | 4.785 |
| ConceptLab | 3.502 | 3.950 |
| C3 | 2.934 | 3.945 |
| VLM-Guided (Ours) | 4.550 | 4.503 |

# E  METRICS AND EVALUATION

## E.1  EVALUATION SETUP

The core idea of our evaluation protocol is to represent images in the CLIP embedding space and compute metrics that characterize the resulting distribution. Standard metrics like the CLIP score measure one-to-one image-text similarity, which is problematic for creativity evaluation — creative outputs should deviate from typical patterns while maintaining category membership. A creative pet that scores lower than a typical cat on CLIP alignment might actually represent a more successful creative generation. Specifically, we use the following metrics: (1) For validity assessment, we employ the CLIP score and GPT-4o verification to ensure outputs remain recognizable as valid category members despite their creative variations. Our goal is not to maximize CLIP score but to remain relatively close to reference values while exploring novel variations; (2) For novelty assessment, we compute relative typicality to measure the difference between broad category similarity (e.g., "pet") and average subcategory similarity (e.g., "cat", "dog"), ensuring outputs avoid conventional modes, alongside GPT-4o Novelty Score which counts how often GPT-4o cannot classify the specific type and responds "unknown"; (3) For diversity assessment, we use distribution-based metrics (total variance and Vendi score (Friedman & Dieng, 2022)) that quantify the spread of creative exploration in the CLIP embedding space.

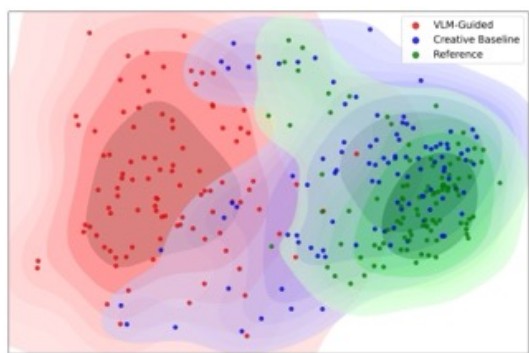

Figure 18: Distribution of fruit CLIP embeddings in 2D PCA space and the Kernel Density Estimation (KDE) of the distributions. Reference images (green): "A photo of a fruit". Creative baseline (blue): "A photo of a new type of fruit". Our VLM-guided method (red): explores diverse regions with minimal overlap with reference.

To evaluate and compare the methods quantitatively, we generate 100 images from four different categories: "pet", "garment", "plant" and "vehicle" using our method, C3, ConceptLab, and two baselines. "Reference" images are generated with SD3.5 from the prompt "A photo of a [category]" and "Creative Prompting" uses the prompt "A photo of a creative / new type of [category]".

## E.2  VISUALIZING THE DISTRIBUTION

We begin by visualizing the resulting distribution in CLIP's space. To do so, we project embeddings to a two dimensional space via PCA. In Figure 18, we visualize the CLIP embedding distributions for "Reference", "Creative Prompting", and our VLM-guided approach. The background distribution is computed on a discrete grid $\mathcal{G}$ of size $50 \times 50$. The density at any point $p \in \mathcal{G}$ is estimated using Gaussian KDE. The plot in Figure 18 shows that our approach pushes mass away from typical exemplars, while the "Creative Prompting" remains close and overlaps with the "Reference" distribution.

### E.3 Novelty and Diversity

To quantify deviation from conventional patterns, we employ two complementary metrics: *Relative Typicality* measures creative deviation from familiar subcategories while maintaining broad category coherence. For a generated image we extract a CLIP embedding $z_i$, using CLIP-ViT-B32, and measure the alignment to the broad category text prompt embedding $t_c$ e.g., "A photo of a pet", and subcategory text prompts embeddings e.g., "A photo of a cat", "A photo of a dog" etc.). Overall, we compute:

$$T_{\text{rel}}(z_i) = \text{cosine\_similarity}(z_i, t_c) - \max_{j \in \{1,\dots,m\}} \text{cosine\_similarity}(z_i, t_s^{(j)}), \qquad (4)$$

where $t_c$ is the CLIP text embedding of the broad category prompt and $\{t_s^{(j)}\}_{j=1}^m$ are the embeddings of subcategory prompts. Positive values indicate the image aligns more with the broad category than with any specific known subcategory, suggesting successful creative generation within the category boundaries.

*GPT Novelty Score* quantifies how often GPT-4o cannot identify the specific type of object. We query GPT-4o to classify each generated image into known subcategories. The score represents the fraction of images classified as "unknown" or unrecognizable variants, directly measuring deviation from familiar modes.

*The Vendi score* (Friedman & Dieng, 2022) quantifies diversity through the Shannon entropy of the eigenvalues of a normalized similarity matrix. Formally, given a collection of samples $x_1, \dots, x_n \in \mathcal{X}$ and a positive semi-definite similarity function $k : \mathcal{X} \times \mathcal{X} \to \mathbb{R}$ with $k(x, x) = 1$, let $K \in \mathbb{R}^{n \times n}$ denote the kernel matrix with $K_{ij} = k(x_i, x_j)$. The Vendi score is defined as:

$$\text{Vendi}(\mathcal{X}) = \exp\left(-\sum_{i=1}^n \lambda_i \log \lambda_i\right) = \exp\left(-\text{tr}\left(\frac{K}{n} \log \frac{K}{n}\right)\right), \qquad (5)$$

where $\lambda_1, \dots, \lambda_n$ are the eigenvalues of $K/n$, with the convention that $0 \log 0 = 0$. This metric can be interpreted as the effective number of dissimilar elements in the sample, ranging from 1 (all identical) to $n$ (all maximally distinct).

*Total Variance*, computed as the trace of the covariance matrix $\text{Tr}(\Sigma) = \sum_{i=1}^d \lambda_i$, measures overall variability across all dimensions in the CLIP embedding space. Higher values indicate greater dispersion and exploration spread.

### E.4 validity

While diversity and novelty distinguish a creative concept from an existing one, validity ensures that it is practical, preventing it from being merely eccentric or nonsensical. We compute the practicality of the generated concepts with two metrics, CLIP text-image alignment score and GPT score to verify semantic validity.

For the GPT score, we provide GPT-4o with a generated image and ask it, "Is this a [category]?". Then we compute the number of times the answer was yes divided by the overall amount of images.

### E.5 Subcategory Selection

For relative typicality computation, we use the following subcategories:
**Pet:** cat, dog, hamster, rabbit, bird, fish, turtle, mouse, gerbil, insect.
**Vehicle:** car, truck, motorcycle, bicycle, bus, train, scooter, van, airplane, drone.
**Plant:** tree, flower, cactus, fern, grass, bush, wildflower, moss, wild mushroom.
**Garment:** shirt, jacket, dress, pants, coat, sweater, hoodie, socks, underwear.

### E.6 VIEScore

VIEScore (Ku et al., 2024) is an explainable automatic metric for evaluating conditional image generation tasks. Instead of relying on similarity scores alone, it uses a Multi-Modal Large Language Model (MLLM, like GPT-4o (OpenAI, 2024)) to produce both a score and a natural language explanation of the judgment. On seven conditional image tasks, VIEScore with GPT-4o reaches a

Spearman correlation of about 0.4 with human ratings (a high correlation value - close to the 0.45 human-to-human agreement). We use VIEScore (Ku et al., 2024) with GPT-4o (OpenAI, 2024) as the base model.

For text-to-image tasks, the metric measures the quality according to two main pillars: first, SC (Semantic Consistency) measures how well the generated image matches the given prompt. This is processed by the MLLM into sub-scores with guiding questions and then combined into a single SC score. Second, PQ (Perceptual Quality) measures how good the image looks visually. It rates things like naturalness, absence of artifacts, distortions, watermarks, and other visual defects, again via sub-scores that are combined into one PQ score. We add an example of the reasoning explanations in Figure 19.

*"A photo of an imaginary pet resting inside a terrarium filled with miniature plants."*

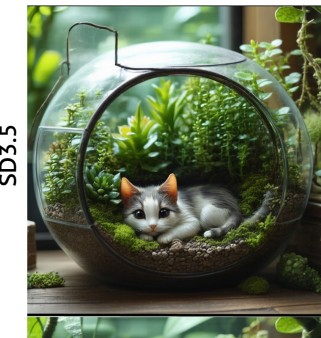

**Semantic Consistency:**
Reasoning: "... it might be argued that a cat could be a common pet and does not heavily emphasize the 'imaginary' aspect. Thus, it slightly misses the unique imaginary characteristic."

**Perceptual Quality:**
Reasoning: "... The image looks very natural overall with the cat comfortably nestled among the plants..."

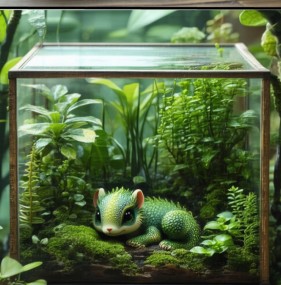

**Semantic Consistency:**
Reasoning: "The image perfectly matches the prompt. It features an imaginary pet ..."

**Perceptual Quality:**
Reasoning: "... The image appears largely natural, with the exception of the context (an animal in a plant terrarium which is an unusual setup)..."

Figure 19: VIE scorer reasoning for controllable creative generation. Top: SD3.5 baseline generates a common cat, receiving lower semantic consistency for missing the "imaginary" aspect. Bottom: Our method produces a more creative creature that better aligns with the "imaginary pet" prompt specification, achieving higher semantic consistency while maintaining perceptual quality.

## F LLM USAGE

Large language models were used exclusively for English language editing and grammatical refinement of the manuscript text. Specifically, we employed LLMs to improve sentence structure, correct grammatical errors, and enhance clarity of technical descriptions. All research ideation, experimental design, implementation, analysis, and scientific conclusions were conducted by the authors. The core technical contributions, methodology, and experimental results represent original work by the authors.

