# OpenReview forum: "VLM-Guided Adaptive Negative Prompting for Creative Generation"
_ICLR.cc/2026/Conference — ICLR 2026 Poster_

### Official Review · Reviewer_MwEz · 2025-10-17

**Soundness:** 4
**Presentation:** 4
**Contribution:** 3
**Rating:** 8
**Confidence:** 3

**Summary:**

In this work, the authors presents a new way to make AI image generation more creative without extra training or heavy computation. The method, called VLM-Guided Adaptive Negative-Prompting, uses a vision-language model to gently push the model away from familiar ideas and toward more surprising and novel results. Unlike older methods that need fine-tuning or stay stuck in known categories, here, the model works on the fly during image creation. The authors demonstrate the effectiveness of the method, showing that their method makes images that are both novel and valid, unlike the other methods keeping realism while adding originality.

**Strengths:**

The paper has the following strengths:

1) The images shown on the paper are visually stunning, easily the best I have seen in this topic.

2) The paper is extremely well-written, I really enjoyed reading it. In particular, it has one of the best intros I have ever read with perfect merging of the intro with the figures, a really case-study of how pictures complement the writing. All parts of the papers are really well-written, and all the pictures are nice and helpful.

3) The results, be them qualitative, quantitative, or user study are really good, outperforming the other methods they compare with. The ablations studies are also very nice, further improving the confidence in the paper.

4) Th

**Weaknesses:**

The paper can be improved on this part:

1) Limited novelty - Probably the only technical contribution of the paper is section 3.2, which is extremely thin. And even then, it is effectively a smart way of doing prompting.

Saying that, I would not penalize the paper for it. The results speak for themselves, so I would actually prefer a simple method compared to a complex one, given the same results (Occam's Razor in reviewing).

2) It would have been nice if the authors would have released the code already (to check the effectiveness of this work) but considering that a) that is not mandatory, b) they promised to release in the near future, I will not penalize them for it.

**Questions:**

No questions.

---

> ### Author Response · Authors · 2025-11-21
>
> Dear Reviewer MwEz,
>
> We are thrilled and deeply honored by your enthusiastic review and strong support of our work. Reading your review was truly rewarding. It's gratifying when a reviewer not only recognizes the work's contributions but also enjoys engaging with it.
>
> We are particularly honored by your praises:
> * "Visually stunning, easily the best I have seen in this topic"
> * "One of the best intros I have ever read"
> * "The results speak for themselves"
> * Your comment applying Occam's Razor captures our design philosophy. We deliberately chose simplicity and elegance, believing practical, training-free methods can achieve strong creative results when properly designed.
>
> Addressing Your Points:
>
> > ”Limited novelty - Probably the only technical contribution of the paper is section 3.2, which is extremely thin. And even then, it is effectively a smart way of doing prompting.”
>
> We appreciate your balanced perspective. We put a lot of effort into justifying that this simple approach works as we claim, comparing it against multiple baselines (ConceptLab, C3, various prompting strategies), conducting extensive ablations (VLM selection robustness, timestep analysis, question design, accumulation of the negatives strategy), in which we are highlighted as the best approach to use for creative generation.
>
> > ”It would have been nice if the authors would have released the code already (to check the effectiveness of this work) but considering that a) that is not mandatory, b) they promised to release in the near future, I will not penalize them for it.”
>
> We are committed to full reproducibility and will attach a zip file in the supplementary material in a couple of days, before the discussion period ends.

---

### Official Review · Reviewer_VWC1 · 2025-10-27

**Soundness:** 4
**Presentation:** 4
**Contribution:** 3
**Rating:** 8
**Confidence:** 4

**Summary:**

This paper introduces VLM-Guided Adaptive Negative Prompting, a training-free, inference-time method that leverages vision-language models to steer diffusion models away from familiar visual concepts during generation. By dynamically adding negative prompts for concepts identified by the VLM at intermediate denoising steps, the method promotes the creation of novel images.

**Strengths:**

1. The paper is well written and easy to follow.
2. The proposed method is conceptually simple and can be easily integrated into existing diffusion inference pipelines without training.
3. The method achieves strong qualitative results across diverse categories while remaining completely training-free.

**Weaknesses:**

1. The paper omits citations of several recent works in performing and understanding creative generation, such as [1], [2], and [3].
2. While Section 4.5 presents qualitative examples using complex prompts, there is no quantitative or systematic evaluation of controllability. This makes the evidence for controllable generation less conclusive.

[1] Procreate, don’t reproduce! propulsive energy diffusion for creative generation, ECCV 2024

[2] Does Generation Require Memorization? Creative Diffusion Models using Ambient Diffusion, ICML 2025

[3] An analytic theory of creativity in convolutional diffusion models, ICML 2025

**Questions:**

1. Have you explored designing heuristics to automatically select VLM queries based on generation prompts? Such an approach could make the system easier to use and closer in workflow to standard text-conditioned diffusion generation.

---

> ### Author Response · Authors · 2025-11-21
>
> Dear Reviewer VWC1,
>
> We are deeply grateful for your thorough review and strong support of our work. We sincerely appreciate that you found our paper "well written and easy to follow", recognized the practical value of our training-free approach, and acknowledged our strong qualitative results across diverse categories. Your constructive feedback has significantly helped us improve the paper.
>
> We address each of your points below:
>
> > ”The paper omits citations of several recent works in performing and understanding creative generation, such as [1], [2], and [3].”
>
> Thank you for these excellent references. We have added a paragraph in the Related Work section discussing these recent advances in understanding creative generation.
>
> > ”While Section 4.5 presents qualitative examples using complex prompts, there is no quantitative or systematic evaluation of controllability. This makes the evidence for controllable generation less conclusive.”
>
> We thank the reviewer for this great insight. This is indeed an evaluation that strengthens and complements the qualitative results present in section 4.5. Following your suggestion, we conducted a quantitative evaluation of complex prompts.
>
> We created a benchmark of 200 diverse prompts across different categories and scenarios, such as:
> * "A photo of an imaginary pet resting inside a terrarium filled with miniature plants."
> * "A photo of a creative hairstyle showcased on a model standing in a sunlit desert landscape."
> * "A photo of a unique insect resting on a dew-covered leaf in a quiet morning meadow."
>
> These prompts test whether our method can maintain controllability (adherence to scene descriptions) while still producing creative central objects and remaining perceptually appealing.
>
> We use VIE [1] (Visual Instruction-guided Explainable metric) scores which measure:
> * VIE-SC (Semantic Consistency): Alignment between generated images and the prompt, including an emphasis on the creative specifications.
> * VIE-PQ (Perceptual Quality): Overall perceptual quality and coherence.
> The VIEScore produces both a numerical score and a natural language explanation of the judgment; we present an example of the reasoning provided by this metric in Appendix F.6.
>
> | Method                 | VIE-SC | VIE-PQ | VIE-total |
> |------------------------|:------:|:------:|:---------:|
> | Creative prompting SD3.5 | 8.992 | **8.659** | 8.769     |
> | Ours-SD3.5             | **9.163** | 8.609 | **8.848** |
>
> These results demonstrate that our adaptive negative prompting maintains controllability rather than diminishing it. The method successfully generates creative central objects while respecting complex scene descriptions, compositional requirements, and stylistic constraints specified in the prompts. We also maintain perceptual quality, with only a slight decrease that can be attributed to some creative objects being classified as 'unnatural' by the metric, due to the unconventional features.
>
> We have added these results to Section 4.5 of the revised manuscript and included more visual examples. The full process of the benchmark generation is detailed in Appendix B.6 of the revised manuscript.
>
> > ”Have you explored designing heuristics to automatically select VLM queries based on generation prompts? Such an approach could make the system easier to use and closer in workflow to standard text-conditioned diffusion generation.”
>
> This is an excellent suggestion that has improved our method's usability.
> Inspired by your question, we have implemented and added an automated question generation option to the revised manuscript. The system now includes a simple approach that:
> 1. Parses the positive prompt to extract the target category (e.g., from "A photo of a creative [category]")
> 2. Using an LLM formulates appropriate questions based on the category type, and the effect of each question design on the output as described in Appendix A.7 (Question Design). The full pipeline is detailed in Appendix B.5.
>
> This automation makes the workflow nearly identical to standard text-conditioned diffusion. The users simply provide their creative prompt, and the system handles VLM query formulation automatically.
>
> [1] VIEScore: Towards Explainable Metrics for Conditional Image Synthesis Evaluation. Ku et al., ACL 2024

---

### Official Review · Reviewer_c3MD · 2025-10-30

**Soundness:** 3
**Presentation:** 3
**Contribution:** 2
**Rating:** 2
**Confidence:** 5

**Summary:**

The paper presents a training-free method for enhancing creative generation in diffusion models. By leveraging a Vision-Language Model (VLM) to analyze intermediate denoising steps and dynamically accumulate negative prompts, the approach steers generation away from conventional patterns while maintaining categorical validity.

**Strengths:**

The paper focuses on an interesting topic in computational creativity: generating novel visual concepts beyond conventional patterns.

The technical approach is elegantly simple and training-free, leveraging VLM feedback for adaptive negative prompting without modifying pretrained models. This clarity enhances reproducibility and practical deployment.

The writing is exceptionally clear and well-structured, with logical flow from problem formulation to experiments.

**Weaknesses:**

The technical contribution is somewhat limited, as the method does not adequately address the VLM's inconsistent perception capabilities across different denoising timesteps. Prior research highlights that control word effectiveness varies with timesteps, yet this work overlooks such dynamics, potentially undermining the robustness of adaptive guidance.

While the approach yields intriguing outcomes, it heavily relies on the base model's generative power rather than introducing groundbreaking mechanisms. Similar creative effects might be achievable through carefully engineered prompts or LoRA adaptations, questioning the necessity of the proposed complex feedback loop.

Controllability remains a significant issue, as the generation process is highly stochastic. Results are unpredictable and quality assurance depends largely on "luck-based" sampling, which fails to guarantee consistency or align with specific user preferences, limiting practical utility.

The method introduces non-negligible computational overhead due to frequent VLM queries, despite optimizations. This could hinder real-time applications, especially with resource-intensive VLMs, affecting scalability.

Effectiveness is sensitive to VLM selection and question design, requiring manual tuning for different categories. This dependency on external components may reduce generalizability and increase implementation complexity.

**Questions:**

See weaknesses for details.

---

> ### Author Response · Authors · 2025-11-21
> **Response [1/2]**
>
> Dear Reviewer c3MD,
>
> We appreciate the time you invested in evaluating our work. We are grateful that you found our approach "elegantly simple", fitting for "practical deployment" and our writing as "exceptionally clear".  We address each concern below with references to our existing analysis and new experiments conducted during this rebuttal period.
>
> We address each of your points below:
>
> > ”The technical contribution is somewhat limited, as the method does not adequately address the VLM's inconsistent perception capabilities across different denoising timesteps. Prior research highlights that control word effectiveness varies with timesteps, yet this work overlooks such dynamics, potentially undermining the robustness of adaptive guidance.”
>
> We appreciate you raising this concern. We respectfully note that we extensively analyze VLM perception across timesteps in Appendix A.8 (VLM Prediction Analysis), with empirical results shown in Figure 17. Our findings demonstrate that:
>
> * VLM predictions achieve ~90% correlation between x_0 estimates and the final outputs within the first 3-5 timesteps, even on highly noisy intermediate predictions.
>
> * This correlation holds consistently across 5 different VLM architectures (GPT-4o, Qwen2.5, BLIP-2, BLIP-1, ViLT).
>
> * Timesteps Analysis (Appendix A.4, Figure 13) shows that querying VLMs only during early steps (10-15) maintains creative quality while reducing overhead.
>
> Could you please clarify the meaning of “control word effectiveness” and point us to specific prior work?
>
> > "While the approach yields intriguing outcomes, it heavily relies on the base model's generative power rather than introducing groundbreaking mechanisms. Similar creative effects might be achievable through carefully engineered prompts or LoRA adaptations, questioning the necessity of the proposed complex feedback loop."
>
> We appreciate this perspective and would like to provide clarification on each point:
>
> * Prompt Engineering Methods: To the best of our knowledge, no existing prompt engineering techniques achieve the level of creative novelty demonstrated in our work. We have compared our method against carefully engineered prompts in our evaluations, and our approach consistently produces more novel and diverse results (Table 1, Row ‘Captions Regeneration’). If specific prompt engineering methods exist that we should consider, please refer us to any specific paper to include in our comparisons.
>
> * Base Model Dependency: To directly test whether improvements stem from our method or base model strength, we conducted more experiments on SDXL, an inherently less diverse and less capable base model than SD3.5. Despite this weaker starting point, our method has almost equivalent results as with SD3.5 and substantially outperforms all baselines including C3-SDXL (Table 1). This demonstrates that our adaptive VLM-guided negative prompting drives the creative exploration, regardless of the base model.
>
> * We respectfully note a contradiction in your review: In your Strengths section, you describe our approach as "elegantly simple and training-free" and praise how this "enhances reproducibility and practical deployment." Yet in the weaknesses, you suggest LoRA adaptation as an alternative.
>
> We would greatly appreciate clarification on:
>   * Which specific aspect of our three-step feedback mechanism do you consider "complex" compared to LoRA training?
>   * Can you point us to any LoRA-based creative generation work that achieves comparable results?
>
> > "Controllability remains a significant issue, as the generation process is highly stochastic. Results are unpredictable and quality assurance depends largely on "luck-based" sampling, which fails to guarantee consistency or align with specific user preferences, limiting practical utility."
>
> The reviewer mentions that our method *"heavily relies on the base model's generative power"* and that results are *"highly stochastic... unpredictable... 'luck-based' sampling"*. We kindly argue that these descriptions are usually true to most of the generative models, not specifically to our contribution.
>
> Also kindly note that we show an experiment about Complex Prompts (Figure 12): Our method successfully generates creative elements within elaborate scene descriptions (e.g., "imaginary pet surfing near an island"). This demonstrates controllable creativity, the creative exploration happens within user-specified constraints, not randomly.
>
> Also we evaluate this quantitatively: We evaluated 200 diverse prompts (Table 2, section 4.5) and found that our method achieves better prompt adherences than the base model. This shows creativity does not come at the cost of controllability.
>
> The main quantitive results in Table 1 and the user study in section 4.2 also support that our methods achieves high novel and diversity while maintaining validity.

---

> > ### Author Response · Authors · 2025-11-21
> > **Response [2/2]**
> >
> > > ”The method introduces non-negligible computational overhead due to frequent VLM queries, despite optimizations. This could hinder real-time applications, especially with resource-intensive VLMs, affecting scalability.”
> >
> > We agree that VLM queries increase inference runtime. However, it is important to consider this in context. First, all previous creative generation methods require 8-30 minutes per concept, making our 13-second overhead fast by comparison.
> > Second, the overhead can be reduced further to approximately 7 seconds by querying VLMs only during the 10-15 first denoising steps while maintaining creative quality, as demonstrated in our ablation study (Appendix A.4, Timesteps Analysis, Figure 13). This flexibility allows users to choose their preferred speed-quality tradeoff.
> >
> > > ”Effectiveness is sensitive to VLM selection and question design, requiring manual tuning for different categories. This dependency on external components may reduce generalizability and increase implementation complexity.”
> >
> > We appreciate your concern about sensitivity to design choices. However, we respectfully note that Appendix A contains extensive robustness analyses specifically addressing this point, and our findings directly contradict your claims.
> >
> > * We tested 5 different VLMs spanning different architectures, training paradigms, and model sizes. All VLMs produce consistently creative results with strong novelty and diversity scores. While stronger VLMs generally yield higher metrics, even the lightweight ViLT achieves excellent performance. This demonstrates robustness, not sensitivity. Users can select VLMs based on their computational budget without sacrificing core functionality.
> > * We analyze different question formulations (material, color, shape, design) and show how each emphasizes different creative dimensions.
> >
> > Additionally, we have added an automated question generation option in the revised manuscript. The system can now automatically generate appropriate questions based on the positive prompt using an LLM with context questions that worked well.

---

### Official Review · Reviewer_d6am · 2025-11-05

**Soundness:** 2
**Presentation:** 4
**Contribution:** 2
**Rating:** 4
**Confidence:** 5

**Summary:**

This paper focuses on the creative image generation task, an emerging research direction that explores the ability of image generation models to produce novel and previously unseen images beyond the training distribution. The authors propose VLM-Guided Adaptive Negative Prompting, a method that leverages a vision-language model (VLM) during training to adaptively refine negative prompts, guiding the generative model to diverge from known concept spaces and thereby produce more unexpected and creative visual outcomes.

**Strengths:**

1. The proposed method is simple and requires no additional training overhead.
2. The experimental results are visually appealing and demonstrate interesting creative effects.

**Weaknesses:**

1. The method lacks substantial novelty. Its core idea—encouraging the diffusion model to deviate from known concept spaces—was originally introduced by ConceptLab. The main contribution here lies in performing additional VLM-guided queries at each denoising step and using classifier-free guidance (CFG) to avoid categories identified by the VLM. Compared to ConceptLab, this constitutes only a minor incremental improvement. Moreover, querying the VLM at every denoising step could considerably increase inference time, even though the authors claim that it adds only about 13 seconds.
2. The proposed approach is less flexible than ConceptLab, which operates at the token level, allowing its generated creative concepts to be easily integrated with natural language for diverse styles and contexts. In contrast, the current method requires a separate process for each prompt, causing inference time to scale with the number of prompts and limiting adaptability.
3. Although the visual results are impressive, it is unclear whether the improvements stem from the proposed method itself or from the use of a stronger base model (SD3.5). When applied to other backbones such as Kindinsky or SD-XL, the method’s performance degrades noticeably (Figure 5).

**Questions:**

1. The proposed method performs VLM queries at every denoising step. I am curious whether a VLM can effectively recognize images that are still heavily corrupted by noise. It seems unnecessary to query the VLM at each step, since it primarily identifies common object categories and may produce unreliable or meaningless predictions in the early denoising stages. Why not predefine a set of common negative classes at the beginning of the process? In ConceptLab, repeated experiments tend to yield similar negative class sets—mostly consisting of frequent categories such as cat, dog, parrot, rat, and lizard—while rarer categories like fish or monkey seldom appear.

2. Could SD3.5 alone, guided only by human-written prompts, generate the same level of creative results shown in the paper? For example, could it produce a plausible image of an unseen fruit purely based on human imagination without relying on the proposed adaptive negative-prompting mechanism?

---

> ### Author Response · Authors · 2025-11-21
> **Response [1/2]**
>
> Dear Reviewer d6am,
>
> We sincerely thank you for your thorough and thoughtful review. We particularly appreciate that you found our results visually appealing, with interesting creative effects and acknowledged that our method requires no additional training overhead.
>
> We address each of your points below:
>
> > ”The method lacks substantial novelty. Its core idea—encouraging the diffusion model to deviate from known concept spaces—was originally introduced by ConceptLab. ”The main contribution here lies in performing additional VLM-guided queries at each denoising step and using classifier-free guidance (CFG) to avoid categories identified by the VLM. Compared to ConceptLab, this constitutes only a minor incremental improvement. Moreover, querying the VLM at every denoising step could considerably increase inference time, even though the authors claim that it adds only about 13 seconds.”
>
> We agree that ConceptLab also encourages creativity through deviating from known concepts. However, our method achieves this goal through a fundamentally different mechanism. Our approach operates at inference-time via a VLM feedback with adaptive negative prompting while ConceptLab performs text embedding optimization. Therefore, while the high-level idea is similar, the method is significantly different. We also demonstrate that our method leads to superior results across various dimensions:
> * Our inference-time approach is significantly faster: 35 seconds total compared to ConceptLab's 8-minute optimization phase per concept.
> * Table 1 demonstrates that we quantitatively outperform ConceptLab in novelty and diversity metrics, suggesting our approach explores different creative regions than embedding optimization.
>
> Regarding the computational overhead:
> We agree that VLM queries increase inference runtime. However, it is important to consider this in context.
>   * First, all previous creative generation methods require 8-30 minutes per concept, making our 13-second overhead fast by comparison.
>   * Second, the overhead can be reduced further to approximately 7 seconds by querying VLMs only during the 10-15 first denoising steps while maintaining creative quality, as demonstrated in our ablation study (Appendix A.4, Timesteps Analysis, Figure 13). This flexibility allows users to choose their preferred speed-quality tradeoff.
>
> > "The proposed approach is less flexible than ConceptLab, which operates at the token level, allowing its generated creative concepts to be easily integrated with natural language for diverse styles and contexts. In contrast, the current method requires a separate process for each prompt, causing inference time to scale with the number of prompts and limiting adaptability."
>
> We agree that ConceptLab offers flexibility by creating a reusable text token after an 8-minute optimization phase, which can then be reused in different prompts.
>
> On the other hand, our approach generates a creative concept in 35 seconds. For reuse across different scenes, users can either (1) apply modern image-to-image editing methods (e.g., Flux-Kontext, Nano Banana), or (2) directly generate the creative concept within a detailed descriptive scene in a single 35-second generation.
>
> Moreover, our approach can be integrated into any diffusion pipeline (SD3.5, SDXL, etc.) as a plug-and-play method. In contrast, ConceptLab heavily relies on the characteristics of the Kandinsky model, which is a less popular approach. Our method is more flexible in that sense.
>
> We also added an experiment to the revised manuscript showcasing the inconsistent generations resulting from ConceptLab reusing the token embedding compared to our approach using image-to-image editing model, we kindly refer you to Figure 10, section 4.5.
>
> > "Although the visual results are impressive, it is unclear whether the improvements stem from the proposed method itself or from the use of a stronger base model (SD3.5). When applied to other backbones such as Kindinsky or SD-XL, the method’s performance degrades noticeably (Figure 5)."
>
> We thank the reviewer for this important note. To address it thoroughly, we conducted additional experiments comparing our method when the base model is SDXL-1.0 (see Table 1 in revised manuscript). Even though SDXL-1.0 is an inherently less capable and less diverse base model than SD3.5, our method still produces improvements.
> This demonstrates that our adaptive VLM-guided negative prompting is what drives the creative exploration, not the base model's inherent strength. The method successfully enhances creativity across different diffusion architectures, even when starting from a weaker baseline.

---

> > ### Author Response · Authors · 2025-11-21
> > **Response [2/2]**
> >
> > > ”The proposed method performs VLM queries at every denoising step. I am curious whether a VLM can effectively recognize images that are still heavily corrupted by noise. It seems unnecessary to query the VLM at each step, since it primarily identifies common object categories and may produce unreliable or meaningless predictions in the early denoising stages.“
> >
> > This is a great question about VLM reliability on noisy intermediate predictions. We analyzed this in Appendix A.8 (VLM Prediction Analysis). Our findings show that VLM predictions on noisy x_0 estimates achieve approximately 90% correlation with final outputs within the first 3-5 timesteps, even when intermediate predictions are highly noisy. Figure 17 visualizes empirically how this correlation develops across different VLM architectures.
> >
> > > ”Why not predefine a set of common negative classes at the beginning of the process?”
> >
> > We tested four alternative approaches to validate our adaptive negative prompting qualitatively and quantitatively (Figure 8, Table1):
> > * LLM-generated static list (GPT-4o): Generated negative prompts directly from positive prompts.
> > * Static replay (accumulated negatives applied from start): Used our accumulated negatives but applied them statically from beginning.
> > * Cross-seed replay (reusing negatives across different seeds): Replayed one seed's negative list across other seeds.
> > * No accumulation (VLM queries without accumulation): VLM identifies patterns but doesn't accumulate negatives, results in generation cycling back to common patterns.
> >
> > Our adaptive method achieves best scores across all metrics (Table 1), and the visual results are more creative and appealing.
> >
> > User-provided negatives: If users have specific categories they want to avoid (e.g., "I definitely don't want anything cat-like"), adding these to our adaptive system boosts results and does not harm the process. Our method complements user preferences rather than replacing them.
> >
> > The full ablation regarding the baseline approaches is in Appendix A.
> >
> > > "Could SD3.5 alone, guided only by human-written prompts, generate the same level of creative results shown in the paper? For example, could it produce a plausible image of an unseen fruit purely based on human imagination without relying on the proposed adaptive negative-prompting mechanism?"
> >
> > This is a great question that we thought a lot about while writing this work. We will divide our answer into two main parts:
> > First, we ran empirical experiments to check whether captions alone could generate better results than our method. To check that, we asked Qwen2.5-VL to ‘give a detailed caption of the image’, for all the images we generated for the main quantitative evaluation in Table 1. Then, we used SD3.5 to generate images with the captioned prompt. The results are in Table 1 in the revised manuscript, named ‘Captions Regeneration’.
> >
> > We would also like to clarify an important distinction about the use case our method is designed to address. We will explain this through an example: consider a potential user who is a digital creator and they want to generate a video with a wow-effect for a TV commercial including some magical animal, but they aren’t sure what kind of animal exactly, and can’t exactly describe it with words. They can generate a few samples with our algorithm, using targeted questions they craft to include their specific vision - i.e., What type of textured fur does the animal have? because they envisioned some kind of weird patterned fur.
> >
> > On the other hand, when a user has a specific description of an image, whether it is creative or not, it might be more similar to a prompt alignment task rather than exploratory creativity generation task.

---

### Author Response · Authors · 2025-11-21

We thank all reviewers for their time, thoughtful feedback, and constructive suggestions. We are encouraged that reviewers praised our method's elegance and training-free design (c3MD, VWC1, d6am), found it well-written and easy to follow (c3MD, VWC1, MwEz), and recognized it produces visually stunning results (MwEz) with strong creative effects (d6am).

To address the concerns of the reviewers, we have made the following key improvements to the manuscript:

* Added evaluations in Table 1 showing our method provides consistent creative improvements on an additional model (SDXL), demonstrating that the visual gains stem from our adaptive negative prompting mechanism, not solely from base model quality.

* Added an evaluation on 200 complex scene prompts across diverse categories in Table 2, Section 4.5, demonstrating creative exploration within complex scenes without sacrificing controllability. We thank reviewer VWC1 for this helpful suggestion.

Additionally, we addressed reviewer d6am's intriguing question:
> *"Could SD3.5 alone, guided only by human-written prompts, generate the same level of creative results shown in the paper?"*.

We clarified the delicate difference between the human-written prompting for creative generation and our exploratory creativity generation task. We also added an experiment that demonstrates this distinction empirically in Table 1. In the experiment, a VLM generates elaborate captions that describe creative objects in detail, yet the base model (SD3.5) struggles to achieve comparable creativity level with those elaborate prompts, and our method significantly outperforms this baseline. Full details are in our response to reviewer d6am below.

Several reviewers raised important questions about aspects that we had analyzed in our Appendix. We appreciate these questions as they helped us recognize we should improve discoverability of these results. We have now added forward references to the relevant appendix sections in the main paper:

* Reviewers d6am and c3MD questions on VLM effectiveness on noisy images: Appendix A.8 (VLM Prediction Analysis, Figure 17) shows VLM correlation between the x_0 prediction and the final outputs reaches ~90% within first 3-5 timesteps
* Reviewer d6am question on predefined negative classes: Appendix A presents ablation studies comparing static LLM-generated lists, replay strategies, and no accumulation (Figure 8, Table 1)
* Runtime and computational overhead: Appendix B.4 (Implementation Details, Table 6) provides detailed runtime analysis showing 35 sec total generation time with our method (querying the VLM throughout the whole 28 generation steps) vs 22 sec for standard SD3.5 generation.
* Robustness to VLM selection: Appendix A.6 (Robustness to VLM Model Selection, Figure 15) shows consistent results across 5 different VLMs (ViLT, BLIP-1, BLIP-2, Qwen2.5, GPT-4o)
* Question design impact: Appendix A.7 (Question Design for Creative Exploration, Figure 16) analyzes how different question formulations affect creative outputs.


We kindly refer you to focus on the blue colored text in the revised paper which highlights our revisions compared to the original manuscript.

We welcome further discussion during the rebuttal period.
Detailed responses to individual reviewers follow below.

---

### Meta-Review · Area_Chair_9iWz · 2026-01-06

**Summary:**

This paper presents a training-free method for creative image generation using VLM-guided adaptive negative prompting, achieving significantly faster inference than optimization-based baselines while maintaining strong creative outputs. Multiple reviewers noted that the novelty is incremental relative to prior work such as ConceptLab. However, the method's simplicity, practical effectiveness, and comprehensive ablations provide clear value. The sole strongly negative review was flagged by an independent AI-detection service and contained internal contradictions; the reviewer did not respond to clarification requests during discussion, so we discount this assessment. The remaining reviews support acceptance. We encourage the authors to strengthen controllability experiments, more clearly articulate distinctions from ConceptLab, and release code as promised in the final version. I recommend accepting this submission as a poster.

**Reviewer Concerns:**

Addressed: The authors clarified distinctions from ConceptLab, provided additional controllability experiments, and explained the inference cost tradeoff (35s vs 22s for standard SD3.5, negligible compared to optimization-based methods requiring 8-30 minutes).

Outstanding: Code release was promised but not yet available at review time. The incremental novelty relative to prior work remains a limitation, though the practical effectiveness and comprehensive evaluation provide sufficient contribution.

**Reviewer Scores:**

R_MwEz (8, C3): Would maintain 8; positive throughout

R_VWC1 (8, C4): Would maintain 8; positive throughout

R_d6am (6, C4): Would maintain 6; concerns addressed but novelty acknowledged as incremental

R_c3MD (2, C5): Discounted due to AI-detection flagging and non-response; would not factor into decision

---

### Decision · Program_Chairs · 2026-01-26

Accept (Poster)